# INTERACT-RAG: REASON AND INTERACT WITH THE CORPUS, BEYOND BLACK-BOX RETRIEVAL

**Yulong Hui**[1], **Chao Chen**[2], **Zhihang Fu**[2], **Yihao Liu**[1], **Jieping Ye**[2], **Huanchen Zhang**[1*]
[1]Tsinghua University    [2]Alibaba Cloud

## ABSTRACT

Retrieval-Augmented Generation (RAG) has significantly enhanced LLMs by incorporating external information. However, prevailing agentic RAG approaches are constrained by a critical limitation: they treat the retrieval process as a black-box querying operation. This confines agents' actions to query issuing, hindering its ability to tackle complex information-seeking tasks. To address this, we introduce Interact-RAG, a new paradigm that elevates the LLM agent from a passive query issuer into an active manipulator of the retrieval process. We dismantle the black-box with a Corpus Interaction Engine, equipping the agent with a set of action primitives for fine-grained control over information retrieval. To further empower the agent on the entire RAG pipeline, we first develop a reasoning-enhanced workflow, which enables both zero-shot execution and the synthesis of interaction trajectories. We then leverage this synthetic data to train a fully autonomous end-to-end agent via Supervised Fine-Tuning (SFT), followed by refinement with Reinforcement Learning (RL). Extensive experiments across six benchmarks demonstrate that Interact-RAG significantly outperforms other advanced methods, validating the efficacy of our reasoning-interaction strategy.

## 1 INTRODUCTION

Large Language Models (LLMs) have shown advancements in natural language understanding and generation but are constrained by their training data, which can be static, outdated, or lack domain-specific knowledge (Huang et al., 2025). Retrieval-Augmented Generation (RAG) has emerged as a prevailing solution to this limitation (Lewis et al., 2020; Gao et al., 2023). By retrieving information from external corpora, RAG systems enable LLMs to access up-to-date information, incorporate specialized knowledge, and reason over proprietary data (Hui et al., 2024; Li et al., 2025f).

The development of RAG has progressed through three stages. The initial approach, **Static RAG**, performs a single retrieval to fetch relevant documents for the LLM (Gao et al., 2023). To handle more complex tasks, **Iterative RAG** frameworks were introduced. These systems employ multi-step retrieval pipelines to progressively gather information (Trivedi et al., 2023; Jiang et al., 2023; Chan et al., 2024). The current frontier is **Agentic RAG**, which uses an LLM-centric agent to autonomously orchestrate the entire workflow with more flexibility (Gao et al., 2025; Li et al., 2025c). The agent decides when to retrieve, what to query, and how to analyze the retrieved information (Singh et al., 2025; Chen et al., 2025). Advanced implementations include prompt-driven multi-agent workflows (Nguyen et al., 2025; Li et al., 2025b) and other end-to-end trained agents using Supervised Fine-Tuning (SFT) and Reinforcement Learning (RL) to improve reasoning and adaptability (Jin et al., 2025a; Zheng et al., 2025; Qian & Liu, 2025).

Despite these advances, existing agentic RAG frameworks share a critical limitation: they treat the retrieval process as an opaque *black-box*. The agent is confined to issuing a query and passively receiving text chunks, typically from an embedding-based semantic retriever (Gao et al., 2025; Jin et al., 2025a). This paradigm prevents the agent from inspecting the internal state of the retrieval process, thereby forcing it to relinquish fine-grained control over the process. Consequently, the agent's exploration is restricted to a trial-and-error loop of query reformulation, which limits the breadth, depth, and overall efficacy of its information seeking. For example, when asked,

---

*Huanchen Zhang is also affiliated with the Shanghai Qi Zhi Institute. Corresponding author.

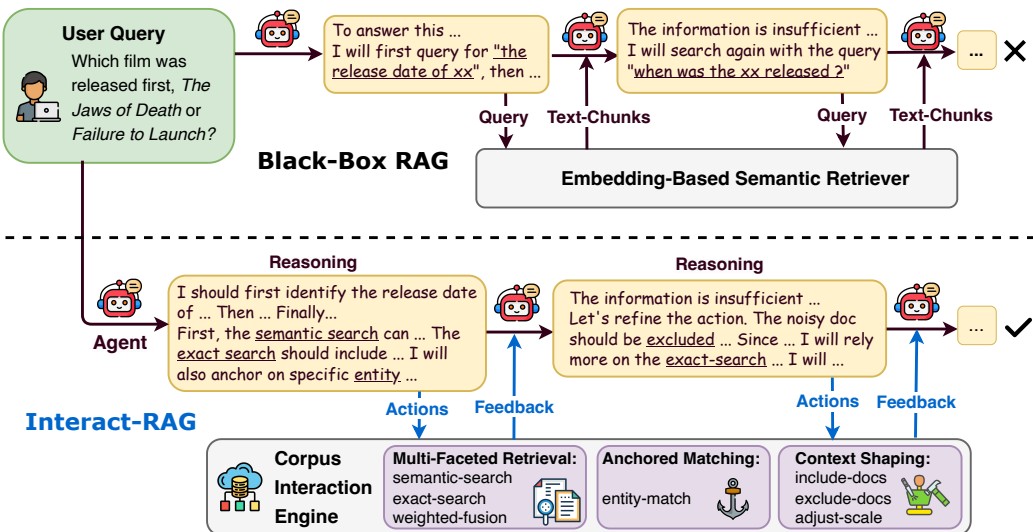

Figure 1: A brief demonstration of Interact-RAG. It empowers the agent with fine-grained control over the information-seeking process, leveraging a set of interactive actions. In contrast, conventional RAG is confined to ineffective loops of query issuing.

"Which film was released first, *The Jaws of Death* or *Failure to Launch*?", an agent might first query for the "release date of *The Jaws of Death*". This retrieval may fail if the supporting evidence is phrased differently (e.g., "...*The Jaws of Death* is a 1976 thriller film...") or if the retriever is distracted by semantically similar but irrelevant entities (e.g., a film named *The Hound of Death*). Faced with such a failure, existing agents can only resort to repeatedly paraphrasing the query (e.g., "when was *The Jaws of Death* released"). This often leads to an inefficient loop of blind guessing that fails to obtain the necessary information.

To overcome this limitation, we introduce **Interact-RAG**, a novel paradigm that transforms the agent from a passive query issuer to an active participant in the retrieval process. Our core idea is to dismantle the retrieval "black box" by providing the agent with transparent and fine-grained control over its information seeking. To achieve this, firstly, we propose a **Corpus Interaction Engine**, which equips the agent with a versatile set of Interaction Primitives, categorized into three action types: (1) Multi-Faceted Retrieval, which allows the agent to employ diverse retrieval strategies (e.g., semantic, exact) and adaptively fuse their results with different weights; (2) Anchored Matching, which focuses the search on a specific entity to mitigate distraction from irrelevant content; (3) Context Shaping, which enables the agent to proactively manage the retrieval context by retaining efficient documents and adjusting the retrieval scope. This suite of primitives serves as a foundation for the fined-grained control, beyond simple query reformulation (as shown in in Figure 1).

However, just providing these interactive capabilities is insufficient. Empowering the LLM to actively and strategically master the interactive pipeline remains challenging. **First**, it is difficult to directly instruct an LLM to manage the intricate multi-step process. To address this, we design a reasoning-enhanced workflow that decomposes the task into three modules: a global planner, an adaptive reasoner, and an executor. This approach not only provides a robust, training-free solution but also synthesizes high-quality agent trajectories for subsequent training. **Second**, achieving full autonomy requires the model to internalize the strategic policies. Therefore, we leverage the synthesized trajectories and apply Supervised Fine-Tuning (SFT), followed by refinement with Reinforcement Learning (RL). As shown in Figure 1, we finally yield a unified, end-to-end agent capable of executing the entire pipeline, without relying on an explicit multi-module architecture.

We conduct extensive experiments on six challenging RAG benchmarks. Our final trained Interact-RAG agent significantly outperforms other advanced RAG approaches, achieving a relative improvement of 22.5%. Ablation studies and detailed analysis further validate the efficacy of our proposed methods. This work sheds light on future exploration to build effective RAG systems with agent-driven interactive retrieval and reasoning enhancement.

## 2 PRELIMINARY

### 2.1 RAG FORMULATION

**External Information.** The external information in RAG is often represented as a visible **corpus** $\mathcal{C} = \{d_1, d_2, \ldots, d_N\}$, typically consisting of $N$ documents or segmented text chunks.

**Task Formulation.** For a RAG system, the core objective is to produce a factual and useful response $A$ to a user query $Q$, utilizing the retrieved information from the external corpus $\mathcal{C}$.

**Basic Pipeline.** The RAG process typically consists of two main stages: retrieval and generation. Given a user query $Q$ and the corpus $\mathcal{C}$, a retriever $\mathcal{R}$ selects some relevant chunks $\mathcal{C}' \subset \mathcal{C}$, which is often based on embedding similarity. Subsequently, a LLM $\mathcal{G}$ generates the response $Y$, conditioned on both the query $Q$ and the retrieved context $\mathcal{C}'$. The process can be formalized as:

$$\mathcal{C}' = \mathcal{R}(Q, \mathcal{C}), \quad Y = \mathcal{G}(Q \mid \mathcal{C}').$$

### 2.2 END-TO-END RAG AGENT

To overcome the rigidity of static pipelines, recent works frame RAG as a sequential process driven by an LLM agent, $\pi_{\text{LLM}}$. Given a query $Q$, the agent continuously searches the information from a corpus $\mathcal{C}$. At each step $t$, it generates an action $a_t$ based on the history: $a_t = \pi_{\text{LLM}}(H_{t-1})$, where the history $H_{t-1}$ contains prior thoughts, actions and retrieved information (with $H_0 = Q$).

The actions of agent often include: (1) `search(`$q_t$`)`: issuing a query $q_t$ to retrieve evidence $I_t$ from the corpus; (2) `answer(`$Y$`)`: concluding the final answer $Y$. When a `search` action is invoked, the information $I_t$ is retrieved and appended to the history, following the action $a_t$:

$$H_t = H_{t-1} \oplus (a_t, I_t)$$

where $\oplus$ denotes the concatenation operation. And a typical agent trajectory can be visualized as:

$$Q \rightarrow [\text{thought}] \rightarrow [\text{search}] \rightarrow [\text{info}] \rightarrow [\text{thought}] \rightarrow [\text{search}] \rightarrow [\text{info}] \rightarrow [\text{thought}] \rightarrow [\text{answer}]$$

In this trajectory, each [thought]-[search] or [thought]-[answer] corresponds to an action $a_t$, [info] represents the retrieved information $I_t$, and their accumulated history $H_t$ is iteratively fed to the LLM for subsequent decisions.

## 3 METHODOLOGY: INTERACT-RAG

In this section, we introduce Interact-RAG with three core components: (1) a corpus interaction engine that supports the fine-grained information control; (2) a reasoning-enhanced workflow that enables both zero-shot solution and data synthesis; and (3) a training pipeline using SFT and RL to produce an autonomous end-to-end agent.

### 3.1 INTERACTIVE ENGINE AND PARADIGM

RAG systems typically treat the information retrieval as a black-box semantic-query-search. To address this, we propose the **Corpus Interaction Engine**, which equips the agent with a versatile set of *Interaction Primitives*. This allows the agent to navigate the information corpus $\mathcal{C}$ in a **human-like manner**, with fine-grained reasoning and manipulation. We define the agent's action space $\mathcal{A}_{\text{CI}}$ (corpus interaction) to include these primitives, which can be categorized into three classes:

**1) Multi-Faceted Retrieval.** Primitives in this category offer diverse retrieval strategies to locate query-related text passages, balancing semantic relevance with lexical precision.

- `semantic_search(`$query_s$`)`: Performs a dense retrieval, using embedding similarity to find semantically related documents.
- `exact_search(`$keywords_e$`)`: Executes a sparse retrieval based on exact keywords ranking, ideal for finding specific terms, names, or phrases.
- `weighted_fusion(`$w_s, w_e$`)`: Sets the fusion weights for semantic and exact search strategies, enabling the agent to flexibly combine their strengths based on the context of the query.

**2) Anchored Matching.** This allows the agent to focus its search on a specific, identified entity, thereby retrieving highly relevant information and minimizing distraction from noisy context.

- `entity_match`($entity$): Retrieves information segments that are strongly associated with a specified entity, ensuring the results are centered around a key subject.

**3) Context Shaping.** These actions enable the agent to sculpt the information context dynamically.

- `include_docs`($doc\_ids$) : Guarantees the inclusion of specified documents in subsequent retrieval steps, ensuring critical information is not missed.
- `exclude_docs`($doc\_ids$): Filters out irrelevant documents from subsequent searches, preventing noisy distractions.
- `adjust_scale`($n$): Adaptively adjusts the scale of the retrieved information (e.g., the number of text chunks) to match the different complexity of the sub-problem.

**Agent Interaction Pipeline.** Within the Interact-RAG pipeline, the LLM agent orchestrates the decision-making process (as shown in Figure 1). At each step $t$, given the previous history, the LLM will generate a structured output that includes: (1) a reasoning thought that rationalizes the current state and strategy, and (2) a suite of concurrent actions $A_t = \{a_{t_1}, a_{t_2}, ...\} \subset \mathcal{A}_{\text{CI}}$. These actions are formulated in the parameterized function call, encapsulated within structured tags (e.g., <tool_call>...</tool_call>). The Corpus Interaction Engine then parses and executes the actions, returning a consolidated response to the LLM. This response, wrapped in tags like <tool_response>, contains the aggregated retrieved content and critical metadata (e.g., source document id, similarity scores for each search strategy). This interactive feedback allows the agent to perform sophisticated strategic analysis and dynamically refine the next actions.

**Implementation.** Our engine is designed for computational efficiency with small overhead. We implement primitives like `exact_search` and `entity_match` by leveraging the Full-Text Search (FTS) modules in relational databases (SQLite, 2025). While this builds an additional text index, the computational cost is negligible. Furthermore, the Context Shaping primitives are implemented through simple filters. It is worth noting that while the agent may invoke multiple strategies in a single iteration, our engine avoids generating multiple large contexts. Instead, it produces a single consolidated context by aggregating these primitives. Further implementation details and validation are provided in Appendix D.2.

### 3.2 REASONING-ENHANCED WORKFLOW

Directly prompting an LLM to master the entire interactive pipeline is challenging. Therefore, we develop a reasoning-enhanced workflow, decomposing the agent action into a hierarchical and iterative structure. It not only serves as a robust training-free solution but also generates high-quality data to train our end-to-end agent. As shown in Figure 2, the workflow contains three collaborative modules: a global-planner, an adaptive-reasoner, and an executor.

**1) Global-Planner.** Given a user query, the global-planner analyzes the problem and decomposes it into a primary step-by-step execution plan, providing a high-level strategic roadmap.

**2) Adaptive-Reasoner.** This component acts as the cognitive core of the workflow. At each step, it first analyzes the current state, including the previous actions, gathered information, and the objective from the planning road-map. After the analysis, it adaptively issues one of two directives:

- **Proceed:** If the current sub-task is progressing well and the retrieved information is sufficient, it instructs the Executor to proceed to the next step or conclude the final response.
- **Reflect & Refine:** If the process encounters an obstacle (e.g., insufficient information), the reasoner will enter a reflection phase. It diagnoses the issue and refines the interaction strategy for the next action. For example, it might rely more on `exact_search` to locate precise terms, or use `exclude_docs` to filter out misleading documents.

Additionally, the reasoner is instructed to adjust the primary plan when necessary. This ensures flexibility, allowing changes without rigidly adhering to the initial roadmap.

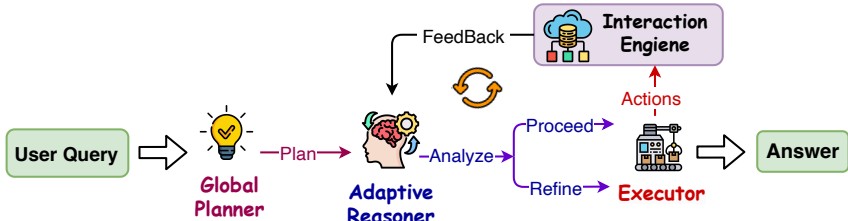

Figure 2: An illustration of our reasoning-enhanced workflow.

**3) Executor.** Following the directives from the reasoner, the executor translate the strategy into a concrete, structured action. It generates the precise function call for the interaction primitives with appropriate parameters. Once all sub-tasks are complete, the Executor will generate the final answer.

This modular design clearly decouples high-level planning, detailed reasoning and precise execution. For a general-purpose LLM, this separation is critical, as the well-defined and focused tasks elicit more reliable output. This workflow yields two significant advantages. First, as a training-free method, it enhances the stability and logical coherence of zero-shot RAG. Second, it serves as a data synthesis engine to train the autonomous agent. With logically-structured modules, the LLM operates in a non-reasoning mode to produce clean reasoning traces, free from the verbose and irrelevant thinking content, which is common in native large reasoning models (LRMs).

## 3.3 END-TO-END AGENT TRAINING

To develop an autonomous, end-to-end LLM agent that internalizes reasoning, we adopt a two-stage training process involving supervised fine-tuning (SFT) followed by reinforcement learning (RL).

**Trace Sampling and Fine-Tuning.** The initial SFT stage aims to teach the LLM the fundamental mechanics, such as planning, reasoning, and mastering the interactions. We leverage our reasoning-enhanced workflow to generate a collection of trajectories based on QA pairs. To ensure the data quality, we retain only successful trajectories, where the agent's final answer matches the ground truth. The agent is then fine-tuned on these high-quality trajectories. The training objective is to predict the sequence of thoughts and actions in an auto-regressive manner. During loss calculation, we mask out the tokens of retrieved information, avoiding the distraction during learning.

**Policy Refinement with Reinforcement Learning.** We then employ RL to enable superior strategies through active exploration. We adopt Group Relative Policy Optimization (GRPO) (Shao et al., 2024), an advanced optimization algorithm, to further refine the agent's policy $\pi_\theta$. During the policy updating, we also mask out the tokens of retrieved information.

**1) RL Objective:** Given a question from the dataset $q \in \mathcal{D}_Q$, the agent generates a group of trajectories $\{\tau_i\}_{i=1}^N$. And the policy $\pi_\theta$ is updated using the following objective function:

$$\mathcal{J}_{\text{GRPO}}(\theta) = \mathbb{E}_{\left[q \sim \mathcal{D}_Q, \, \{\tau_i\}_{i=1}^N \sim \pi_{\theta_{\text{old}}}(\cdot|q)\right]}$$

$$\left[ \frac{1}{N} \sum_{i=1}^N \frac{1}{|\tau_i|} \sum_{t=1}^{|\tau_i|} \min \left( \rho_\theta(\mathbf{a}_t^{(i)}) \hat{A}(\tau_i), \text{clip}\left( \rho_\theta(\mathbf{a}_t^{(i)}), 1 \pm \epsilon \right) \hat{A}(\tau_i) \right) - \beta \, \mathbb{D}_{\text{KL}}(\pi_\theta \| \pi_{\text{ref}}) \right],$$

where $\mathbf{a}_t$ means the agent action, $\rho_\theta(\mathbf{a}_t^{(i)}) = \frac{\pi_\theta(\mathbf{a}_t^{(i)}|\mathbf{s}_{t-1}^{(i)})}{\pi_{\theta_{\text{old}}}(\mathbf{a}_t^{(i)}|\mathbf{s}_{t-1}^{(i)})}$ is the importance sampling ratio, and the

advantage $\hat{A}(\tau_i)$ is calculated by normalizing the rewards within the sampled group. This objective encourages updates towards high-reward trajectories while stabilizing training.

**2) Reward Function:** We design a outcome reward $R(\tau)$ to guide the agent, based on both the syntactic validity and answer accuracy of its trajectory $\tau$:

$$R(\tau) = -1 + \mathbb{I}\{\tau_{\text{valid}}\} + \mathbb{I}\{\tau_{\text{valid}}\} \cdot \mathbb{I}\{y_{\text{ans}}\}$$

Here, each trajectory incurs an initial penalty of -1. The agent should generate a format-coherent output to overcome this penalty. $\mathbb{I}\{\cdot\}$ denotes the *indicator function*, which returns 1 if its enclosed

condition is true, and 0 otherwise. **First**, the term $\mathbb{I}\{\tau_{\text{valid}}\}$, grants a +1 reward if $\tau$ is syntactically valid, thereby neutralizing the initial penalty. Syntactic validity encompasses the entire action sequence structure, the reasoning format, and the tool call syntax. **Second**, $\mathbb{I}\{\tau_{\text{valid}}\} \cdot \mathbb{I}\{y_{\text{ans}}\}$ provides a +1 reward for task success, where the final answer $y_{\text{ans}}$ matches the ground-truth. This reward is gated by the trajectory's validity, ensuring that only well-formed output can be rewarded. It is worth noting that, the inclusion of the "-1 initial penalty" was primarily intended for intuitive logic. Mathematically, this constant term is canceled out during the group-based normalization process in GRPO and does not influence the advantages or the actual training.

## 4 EXPERIMENTS

### 4.1 EXPERIMENTAL SETTINGS

**Datasets.** We conduct experiments across six prominent and standard RAG benchmarks. These include two single-hop question-answering datasets, Natural Questions (**NQ**) (Kwiatkowski et al., 2019) and **PopQA** (Mallen et al., 2023), and four multi-hop question-answering datasets: **HotpotQA** (Yang et al., 2018), 2WikiMultiHopQA (**2Wiki**) (Ho et al., 2020), **MuSiQue** (Trivedi et al., 2022), and **Bamboogle** (Press et al.). More dataset details are in Appendix D.1 Besides, for more comprehensive evaluation over non-wiki domains, we conduct additional experiments on the MultiHop-RAG benchmark (Tang & Yang, 2024) (more details in Appendix C.2).

**Baselines.** We compare our method against a diverse suite of baselines, covering paradigms of non-RAG, static, iterative, prompt-driven multi-agent, and end-to-end trained agents. Specifically, we include: (1) **Direct**: Answers questions directly via Chain-of-Thought, without external information. (2) **Standard RAG**: A static RAG method that performs a single retrieval. (3) **IR-CoT** (Trivedi et al., 2023): A representative iterative RAG method using intermediate thought-chain steps to formulate queries for multi-step retrieval. (4) **MA-RAG** (Nguyen et al., 2025): A multi-agent framework with agent collaboration. (5) **Search-O1** (Li et al., 2025b): An agentic framework with a reasoning-enhanced workflow. (6) **Search-R1** (Jin et al., 2025a): An end-to-end approach that uses RL to generate multi-turn search queries after reasoning. (7) SimpleDeepSearcher (**S-DeepSearcher**) (Sun et al., 2025): An end-to-end approach that fine-tunes a LLM on synthesized high-quality data. (8) **R-Search** (Zhao et al., 2025): An end-to-end approach that trains an autonomous agent via RL, using optimized multi-reward signals.

**Experimental Details.** Following previous works (Jin et al., 2025a; Qian & Liu, 2025), we process the 2018 Wikipedia dump as the retrieval corpus. We employ the e5-base-v2 (Wang et al., 2022) model as the retriever, fetching the top 3 relevant chunks. To ensure a fair comparison, we maintain the same corpus and retriever across all methods, and re-evaluate all baselines under this unified setting. For all experiments, we use Qwen3-8B by default (Yang et al., 2025), a recent instruction-tuned model. For training-driven baselines (i.e., Search-R1, S-DeepSearcher, and R-Search), we utilize their official checkpoints trained on Qwen-2.5-7B, since their 8B versions are not available yet. To ensure the comprehensiveness, we also report our results on Qwen2.5-7B in main results.

We train our agent on the combined training splits of NQ, HotpotQA, and MuSiQue, and evaluate it on the test splits of all six benchmarks. This setup enables the generalization on both **in-distribution** and **out-of-distribution** (PopQA, 2Wiki, Bamboogle). For the training process, we first employed Qwen-Plus to synthesize 4.8K agent trajectories for SFT. Subsequently, we utilized 7.1K question-answer pairs for the RL phase. More details are in Appendix D.3.

### 4.2 MAIN RESULTS

We evaluate Interact-RAG on six benchmarks, with the main results in Table **??**. Our findings highlight three key advantages of our approach. **First,** Interact-RAG consistently achieves best performance across all datasets. On average, it improves the EM-score by **9.7** points (**22.5%** relative gain) over the second-best method, Search-R1. And our 7B version also achieves a relative improvement of 10.1%. Notably, our Interact-RAG was trained on 12K QA data, a small fraction of the 170K QA pairs used for Search-R1. This data disparity also explains Search-R1's higher results on the NQ dataset using the 7B model. **Second,** the performance gains are more pronounced on complex multi-hop QA tasks. For instance, on Musique, Interact-RAG delivers a 36.4% relative improvement. Concurrently, it maintains strong performance on single-hop benchmarks like

| Method | Multi-Hop QA | | | | | | | | Single-Hop QA | | | | AVG | |
| | HotpotQA | | 2Wiki. | | Musique | | Bamboogle | | NQ | | PopQA | | | |
| | EM | F1 | EM | F1 | EM | F1 | EM | F1 | EM | F1 | EM | F1 | EM | F1 |
|---|---|---|---|---|---|---|---|---|---|---|---|---|---|---|
| *Results on Qwen2.5-7B Backbone* | | | | | | | | | | | | | | |
| Direct | 17.8 | 26.2 | 22.6 | 27.7 | 4.3 | 9.8 | 19.6 | 30.2 | 16.1 | 23.5 | 18.6 | 21.2 | 16.5 | 23.1 |
| Std-RAG | 29.8 | 41.5 | 28.0 | 34.1 | 9.5 | 14.4 | 18.4 | 26.8 | 35.1 | 44.9 | 34.6 | 41.2 | 25.9 | 33.8 |
| IR-CoT | 19.3 | 36.6 | 20.0 | 37.5 | 5.2 | 14.4 | 18.2 | 30.0 | 16.5 | 27.5 | 24.6 | 34.8 | 17.3 | 30.1 |
| MA-RAG | 35.0 | 44.7 | 39.6 | 46.9 | 13.1 | 19.0 | 40.8 | 50.9 | 29.5 | 39.9 | 33.3 | 39.0 | 31.9 | 40.1 |
| Search-o1 | 33.6 | 46.3 | 39.9 | 49.6 | 14.7 | 21.7 | 32.0 | 45.3 | 33.7 | 43.8 | 36.3 | 43.1 | 31.7 | 41.6 |
| Search-R1 | 45.2 | 60.1 | 50.9 | 58.2 | 25.5 | 34.1 | 42.2 | 55.6 | **45.3** | **54.7** | 49.3 | 53.6 | 43.1 | 52.7 |
| R-Search | 38.2 | 51.0 | 58.8 | 64.4 | 19.4 | 28.0 | 36.0 | 52.1 | 36.8 | 46.5 | 42.8 | 46.2 | 38.7 | 48.0 |
| S-DeepSearch | 40.2 | 53.6 | 54.0 | 61.8 | 18.6 | 25.8 | 46.2 | 57.4 | 37.0 | 46.6 | 40.6 | 46.1 | 39.4 | 48.5 |
| Interact-RAG-7B | **47.8** | **61.6** | **63.6** | **71.0** | **30.9** | **39.5** | **47.6** | **61.1** | 43.7 | 52.9 | **51.6** | **54.4** | **47.5** | **56.8** |
| *Results on Qwen3-8B Backbone* | | | | | | | | | | | | | | |
| Direct | 23.6 | 33.0 | 26.4 | 32.5 | 5.9 | 13.3 | 33.2 | 49.1 | 19.8 | 30.4 | 20.5 | 25.0 | 21.6 | 30.6 |
| Std-RAG | 37.6 | 50.6 | 35.9 | 41.2 | 13.7 | 21.0 | 26.8 | 35.0 | 37.1 | 47.8 | 37.3 | 44.6 | 31.4 | 40.0 |
| IR-CoT | 30.8 | 43.3 | 33.6 | 42.6 | 12.9 | 20.3 | 22.8 | 32.6 | 33.0 | 43.9 | 31.7 | 38.1 | 27.5 | 36.8 |
| MA-RAG | 39.3 | 51.7 | 45.5 | 53.1 | 18.0 | 25.1 | 35.6 | 49.3 | 34.6 | 46.2 | 40.2 | 45.8 | 35.5 | 45.2 |
| Search-o1 | 23.1 | 30.2 | 28.0 | 34.6 | 10.1 | 13.8 | 31.2 | 39.7 | 33.1 | 41.9 | 33.1 | 38.4 | 26.4 | 33.1 |
| Search-R1[†] | 45.2 | 60.1 | 50.9 | 58.2 | 25.5 | 34.1 | 42.2 | 55.6 | 45.3 | 54.7 | 49.3 | 53.6 | 43.1 | 52.7 |
| R-Search[†] | 38.2 | 51.0 | 58.8 | 64.4 | 19.4 | 28.0 | 36.0 | 52.1 | 36.8 | 46.5 | 42.8 | 46.2 | 38.7 | 48.0 |
| S-DeepSearch[†] | 40.2 | 53.6 | 54.0 | 61.8 | 18.6 | 25.8 | 46.2 | 57.4 | 37.0 | 46.6 | 40.6 | 46.1 | 39.4 | 48.5 |
| Interact-RAG | **51.6** | **66.7** | **69.6** | **76.4** | **34.8** | **43.9** | **54.0** | **65.5** | **50.9** | **60.7** | **56.0** | **60.2** | **52.8** | **62.2** |

Table 1: Overall performance in Exact Match (EM) and F1 scores across various benchmarks. **Bold** and underline denote the best and second-best performance. For the Qwen3-8B results, methods marked with a dagger ([†]) use their official 7B models, due to the lack of 8B versions. Notably, our Interact-RAG was trained on 12K QA data, a small fraction of the 170K QA pairs used for Search-R1. This data disparity also explains Search-R1's higher results on the NQ dataset when using the 7B model.

| Method | 2Wiki. | Musique | PopQA |
|---|---|---|---|
| Interact-RAG | **69.6** | **34.8** | **56.0** |
| w/o Interaction | 63.4 (-8.9%) | 30.1 (-10.9%) | 50.2 (-10.4%) |
| w/o SFT | 59.0 (-15.2%) | 26.4 (-21.9%) | 52.2 (-6.8%) |
| w/o RL | 65.2 (-6.3%) | 28.1 (-16.9%) | 45.6 (-18.6%) |

Table 2: Ablation study on Interact-RAG, reported in Exact Match (EM) scores. The 2Wiki and Musique are multi-hop-QA datasets, while PopQA is single-hop.

NQ and PopQA, with relative improvements of 11.0% and 12.3% on EM scores. This validates the effectiveness of our interaction-reasoning paradigm in tackling complex challenges. **Third,** our trained agent demonstrates great generalization. Trained with train-splits of HotpotQA, Musique, and NQ, it achieves consistent improvements on both in-distribution and out-of-distribution benchmarks. This indicates that the learned capability are not task-specific, underscoring the robustness and generalizability of our approach.

## 4.3 ABLATION STUDY

As shown in Table 2, we conduct an ablation study on Interact-RAG.

**Efficacy of the Interaction Paradigm.** The "w/o Interaction" variant means the black-box retrieval is deployed, mirroring the paradigm of typical agentic RAG systems. In this configuration, the agent is restricted to issuing queries to a semantic retriever, without any other interaction. The

| Method | 2Wiki | Musique | PopQA |
|---|---|---|---|
| Interact-RAG | **69.6** | **34.8** | **56.0** |
| w/o All-Interaction | 63.4 | 30.1 | 50.2 |
| w/o Multi-faceted Retrieval | 66.0 | 34.6 | 55.1 |
| w/o Anchored Matching | 66.3 | 34.4 | 53.4 |
| w/o Context Shaping | 68.8 | 33.6 | 55.2 |

Table 3: Fine-grained ablation study on interaction primitives, reported in Exact Match (EM) scores.

| Training-free Method | 2Wiki. | Musique | PopQA |
|---|---|---|---|
| MA-RAG | 45.5 | 18.0 | 40.2 |
| Interact-RAG-Workflow | **60.1** | **24.1** | **43.6** |
| w/o Interaction | 56.3 | 18.8 | 38.7 |
| w/o Workflow | 52.0 | 21.8 | 40.0 |

Table 4: Ablation performance of our training-free workflow, with MA-RAG (Nguyen et al., 2025) as a baseline reference. Results are reported in Exact Match (EM) scores.

corresponding results clearly show a marked performance drop. This finding underscores the critical value of our interactive paradigm, confirming that equipping the agent with fine-grained control over the information-seeking process is essential and effective.

**Impact of the Training Strategy.** For our two-phase training, removing SFT leads to severe performance drops, especially on challenging datasets like Musique (-21.9%). This highlights its role in building fundamental mechanics of planning, reasoning, and iterative interaction. Similarly, omitting RL also causes marked declines, as RL is essential to develop more strategic policies. These results demonstrate that while SFT establishes the core patterns of reasoning and interaction, RL further optimizes the agent's policy to achieve better performance. (More discussion in Section 4.5).

**Ablation on Interaction Primitives.** To isolate the specific contributions of each component, we conduct a fine-grained ablation study on the interaction primitives (Table 4.3). The results reveal a strong synergistic effect: while removing the single module degrades performance, the complete absence of interaction leads to the most significant drop. This confirms that these primitives function collectively to surpass black-box retrieval. Furthermore, the contributions of different components vary across data patterns, showing the comprehensiveness of our design. For example, Multi-Faceted Retrieval and Anchored Matching are vital for 2Wiki, which may demand explicit facts. And Context Shaping proves more impactful on Musique, where the prevalence of distractors requires robust context scaling.

## 4.4 TRAINING-FREE SCENARIOS

In scenarios with limited training resources or requiring zero-shot deployment, training-free solutions are practically important. Therefore, we evaluate our training-free approach, termed Interact-RAG-Workflow. As shown in Table 4, our approach consistently outperforms MA-RAG across various benchmarks, underscoring the intrinsic effectiveness of our reasoning-interaction paradigm even without model training. To better understand the impact of individual components, we conduct two ablation studies. First, removing the interaction (i.e, resorting to a black-box query-search) leads to a significant performance drop, highlighting the critical role of fine-grained retrieval control. Second, "w/o workflow" means omit our reasoning-enhanced workflow and directly instruct the LLM through an end-to-end prompt (detailed in Appendix D.4). This also results in performance degradation, confirming the effectiveness of our workflow to orchestrate the entire RAG process.

## 4.5 DETAILED ANALYSIS

**Efficiency of Information Retrieval.** We further assess the retrieval process by measuring the number of action iterations. We compare our Interact-RAG against two query-only methods: an ablation variant restricted to only the query-search action (termed as Ours-Search) and the Search-

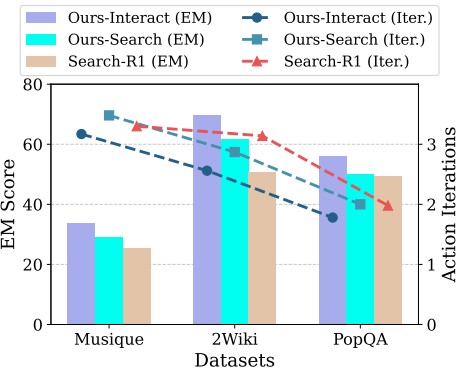

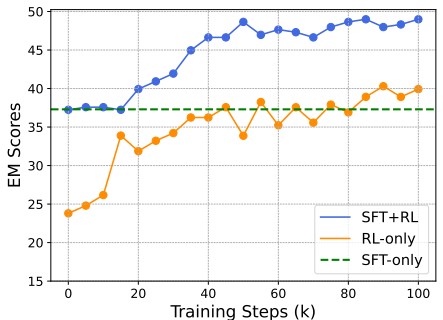

Figure 3: Comparison of the retrieval iterations and the accuracy.

Figure 4: Performance during RL training. Measured on a sampled subset.

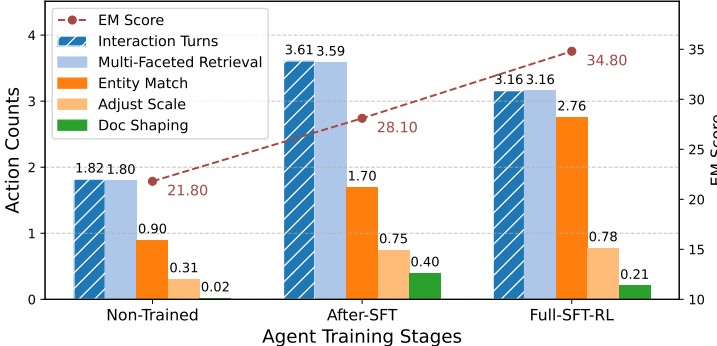

Figure 5: Action invocation status in different training stages. Measured on the Musique dataset.

R1 method. The results in Figure 3 indicate that Interact-RAG always achieves the highest EM scores with the minimum action iterations. This is particularly pronounced on complex multi-hop datasets (2Wiki and Musique), where tasks demand more intricate information seeking. This finding validates the core advantage of our paradigm: by providing the agent with fine-grained control, it can navigate the information space in less iterations, avoiding inefficient trial-and-error loops. (A case study is in Figure 6).

**Training Dynamics in RL.** Figure 4 depicts the RL training dynamics, with EM scores evaluated on a sampled subset of the six test datasets. We compare the two-stage SFT+RL approach with a RL-only method. Starting from the SFT checkpoint, the SFT+RL model demonstrates a consistent improvement after an initial warm-up phase, ultimately converging at a high-performance level. In contrast, the RL-only agent shows faster progress within the first 40 steps and then its development slows, resulting in marginal improvements over the SFT-only baseline (dashed line) and falling significantly behind the SFT+RL model. This highlights the critical role of the two-stage training. SFT provides the agent with a crucial foundational capability and strategic solution paths. Without this prior, the RL-only agent struggles to master the complex retrieval strategies from scratch.

**Interaction Patterns Across Training Stages.** To understand how our training shapes the agent's behavior, Figure 5 shows the statistics of interaction across different stages. (1) Non-Trained: The agent relies solely on an end-to-end prompt, exhibiting limited engagement. It averages only 1.82 turns, with minimal invocation of interactive actions. This confirms that, without training, the LLM struggles to autonomously master the iterative information-seeking process. (2) SFT Stage: After SFT, the agent learns the fundamental processing patterns. The number of interaction turns rises to 3.61, indicating that SFT instills reasoning strategies and equips the agent to better engage with the Corpus Interaction Engine. (3) RL Stage: While the number of interaction turns decreases, the EM score improves significantly. This reflects the agent's transition to a more strategic policy, enhancing both efficiency and accuracy through improved reasoning and appropriate retrieval actions. (4)

Detailed Observations: After the RL exploration, the frequency of Entity-Match increases sharply. This suggests the agent has learned to prioritize precise and anchored searches. In contrast, the use of Doc-Shaping decreases, because the agent's improved retrieval precision reduces the necessity for subsequent noise filtering. In summary, this progression analysis highlights the rationality and effectiveness of our interaction paradigm and training pipeline.

## 5 RELATED WORK

### 5.1 RETRIEVAL-AUGMENTED GENERATION

Retrieval-Augmented Generation (RAG) is a prevailing method to enhance LLMs with external information (Lewis et al., 2020). Basic RAG relies on static embedding-based retrieval (Lewis et al., 2020; Li et al., 2025e;d), which may suffer from information omission (Gao et al., 2023). To address this, various studies propose tree-based or graph-based index to improve retrieval robustness (Jin et al., 2025b; Edge et al., 2024; Luo et al., 2025). Another direction focuses on improving the retrieval pipeline. Iterative RAGs were introduced to progressively refine information through multi-step retrieval (Trivedi et al., 2023; Chan et al., 2024; Hui et al., 2025). Recent agentic methods provide more flexibility, where the LLM autonomously orchestrates the entire RAG pipeline (Gao et al., 2025). Methods such as MA-RAG (Nguyen et al., 2025), Search-O1 (Li et al., 2025b), and MCTS-RAG (Hu et al., 2025) implement prompt-driven strategies, leveraging multiple agentic modules. End-to-end approaches like Search-R1 (Jin et al., 2025a), InForage (Qian & Liu, 2025), and SimpleDeepSearcher (Sun et al., 2025) adopt SFT and RL to create fully autonomous agents. Despite the effectiveness of above approaches, they often operate within a black-box retrieval paradigm, limiting the analysis and control. Addressing this, our work explores an interactive framework with fine-grained retrieval manipulation, supporting improved reasoning and adaptability.

### 5.2 REASONING-ENHANCED LLM AGENT

Enhancing LLMs with reasoning has become a prevailing research focus (Xu et al., 2025). The strategies span prompting-based approaches like Chain-of-Thought Wei et al. (2022), and training-optimized models like OpenAI o1/o3/o4 (Jaech et al., 2024) and DeepSeek-R1 (Guo et al., 2025). To support broader scenarios, various works leverage reasoning to improve the performance of LLM agents(Ferrag et al., 2025), training them to use tools and solve complex problems (Lu et al., 2025b; Shen et al., 2025). They explore various dimensions, including the construction of high-quality training data (Li et al., 2025a; Shi et al., 2025), the refinement of reward signals (Zhao et al., 2025; Qian & Liu, 2025), and the optimization of reinforcement learning algorithms (Dong et al., 2025; Lu et al., 2025a). While these works have made great advances concentrating on the agent's training, our focus is distinct: we redesign the interaction paradigm for RAG agents and leverage the reasoning capability to enable fine-grained manipulation.

## 6 CONCLUSION

In this paper, we identify the limitation of simple black-box retrieval, and introduce Interact-RAG, a new paradigm empowering LLM agents with fine-grained control over the information-seeking process. Our approach features an underlying Interaction Engine, a reasoning-enhanced workflow and a two-stage training pipeline, finally yielding a unified, end-to-end interactive RAG agent. Extensive experiments show Interact-RAG significantly outperforms advanced baselines, validating the effectiveness of reasoning-interaction paradigm. This work offers a promising direction for creating more powerful, transparent, and interactive RAG systems.

ETHICS STATEMENT

This work adheres to the ethical guidelines set forth by ICLR 2026. We have conducted our research with a commitment to avoiding harm, ensuring honesty and transparency in our methodology and reporting. We have made concerted efforts to identify and mitigate potential biases in our data and algorithms to ensure fairness. Furthermore, our research respects individual privacy, and we have complied with all applicable regulations and ethical standards regarding data use.

REPRODUCIBILITY STATEMENT

In this work, we present Interact-RAG, a new paradigm empowering LLM agents with fine-grained control over the information-seeking process. To ensure reproducibility and facilitate further research, we provide the source code in the supplementary materials. Additionally, we offer comprehensive documentation in Section 4.1 and Appendix D, covering dataset details, training parameters, environment constructions, and experimental configurations.

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

## A  THE USE OF LLMs

This paper utilized LLMs only for language polishing in parts of the text.

## B  LIMITATIONS

Despite the effectiveness of Interact-RAG, we acknowledge two limitations: (1) RL Optimization: Our RL stage currently relies on outcome-based rewards. While this aligns with some established GRPO practices, incorporating granular process rewards or advanced RL algorithm could further enhance learning efficiency. (2) Test-time Cost: Our explicit Chain-of-Thought reasoning trajectory incurs higher latency and token usage, compared to non-reasoning modes. This test-time scaling reflects the trade-off inherent in reasoning models, where increased compute is exchanged for superior accuracy. In future work, we will explore more advanced RL designs and optimize the cost-effectiveness.

## C  ADDITIONAL DEMONSTRATION AND EXPERIMENTS

### C.1  CASE STUDY

As illustrated in Figure 6, Search-R1, which relies on black-box query search, can fall into **query loops**, hindering its ability to efficiently retrieve evidence. In contrast, our Interact-RAG utilizes granular interactive actions to effectively address this challenge.

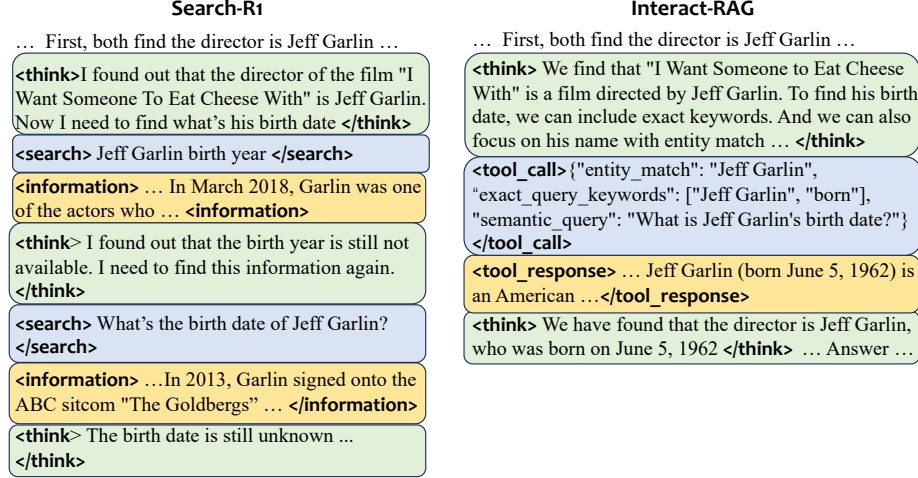

Figure 6: Case study under a multi-hop query, comparing Interact-RAG and Search-R1. The results highlight that Search-R1, relying on black-box query-search, can become trapped in **query loops**, failing to retrieve efficient evidence. In contrast, our approach leverages **granular interactive actions** to directly resolve the issue. Both the exact keyword "born" and the anchored entity-match are helpful to retrieval the desired information.

### C.2  GENERALIZATION TO OTHER DOMAINS

For more comprehensive evaluation over other non-wiki domains, we conducted additional experiments on the MultiHop-RAG benchmark (Tang & Yang, 2024). Its corpus consists of news covering diverse fields such as technology, health, and business. As shown in the Table 5, Interact-RAG significantly outperforms baselines in this out-of-distribution setting. This further demonstrates the effectiveness and robustness of our interaction-reasoning paradigm.

| Method | EM | F1 |
|---|---|---|
| *7B Models* | | |
| Std-RAG-7B | 57.5 | 59.4 |
| MA-RAG-7B | 50.2 | 52.3 |
| Search-R1-7B | 73.0 | 74.1 |
| Interact-RAG-7B | **79.4** | **80.4** |
| *8B Models* | | |
| Std-RAG-8B | 63.4 | 65.1 |
| MA-RAG-8B | 66.3 | 68.7 |
| Interact-RAG-8B | **81.4** | **82.3** |

Table 5: Performance results on the MultiHop-RAG benchmark (non-wiki domain).

| Method | 2Wiki | Musique | HotpotQA |
|---|---|---|---|
| Standard-Retrieval | 0.434 | 0.127 | 0.502 |
| MA-RAG-7B | 0.620 | 0.256 | 0.573 |
| MA-RAG-8B | 0.631 | 0.277 | 0.581 |
| Search-R1-7B | 0.608 | 0.256 | 0.609 |
| Interact-RAG-7B | 0.686 | 0.316 | 0.632 |
| Interact-RAG-8B | **0.706** | **0.335** | **0.637** |

Table 6: Retrieval quality evaluation. We report the coverage of supporting facts within the retrieved context.

## C.3 RETRIEVAL QUALITY

To assess retrieval quality beyond final answer accuracy, we evaluate the coverage of supporting facts across three multi-hop benchmarks, since they provide human-annotated evidence. We calculated the proportion of supporting facts covered within the retrieved context. As shown in Table 6, our Interact-RAG consistently outperforms other baselines. This confirms that our active interaction paradigm effectively improves the information retrieval and gathering process.

## C.4 COST-PERFORMANCE TRADE-OFF ANALYSIS.

Table C.4 provides a detailed comparison of computational costs on the 2WikiMultiHopQA dataset. The evaluation was conducted on the NVIDIA A100 GPU utilizing the vLLM inference engine, and the results are averaged per question. To ensure the consistent comparison, the methods are all based on the Qwen2.5-7B model. We acknowledge that Interact-RAG incurs higher latency and token consumption compared to standard baselines. This increase stems from the explicit reasoning trajectory. However, this design aligns with the emerging paradigm of reasoning models and test-time scaling (e.g., OpenAI-o1 (Jaech et al., 2024), DeepSeek-R1 (Guo et al., 2025)), where increased inference compute is a necessary trade-off for superior accuracy over complex tasks.
Furthermore, we conducted a control experiment applying a Best-of-N ($N = 3$) sampling strategy to the strong baseline, Search-R1. While this approach increased the computational cost by over $3\times$, the performance only improved from 50.9 to 56.2, still falling short of ours. This demonstrates the effectiveness of Interact-RAG.

## C.5 FUSION POLICY FOR MULTI-FACETED RETRIEVAL

In the multi-faceted retrieval, we need to fuse the results from two distinct strategies. In our implementation, we apply min-max normalization to both scores, and then perform the weighted aggregation, where the weight is assigned by the LLM agent. Now, we compare our fusion policy with two standard rank fusion methods, including Reciprocal Rank Fusion (RRF) and CombMNZ (Cormack et al., 2009). As shown in Table 8 , our implementation yields performance comparable or slightly

| Method | EM Score | Latency (s) | Tool Time (s) | Out Tok. | In Tok. |
|---|---|---|---|---|---|
| MA-RAG-7B | 39.6 | 15.1 | 5.09 | 181 | 3812 |
| Search-R1-7B | 50.9 | 17.2 | 5.61 | 192 | 4289 |
| Interact-RAG-7B | 63.6 | 36.4 | 4.67 | 713 | 5216 |

Table 7: Detailed computational cost analysis on the 2WikiMultiHopQA dataset. We report the Exact Match (EM) score alongside average wall-clock latency, tool execution time, and accumulated token consumption per query.

| Fusion Method | 2Wiki | Musique | PopQA |
|---|---|---|---|
| RRF | 68.7 | 34.6 | 55.3 |
| CombMNZ | 69.8 | 32.9 | 56.0 |
| Interact-RAG | 69.6 | 34.8 | 56.0 |

Table 8: Comparison of different fusion strategies (reported in EM Scores).

superior. And we would like to clarify that the fusion mechanism is a small functional design in our interaction engine, so its variations do not significantly alter the overall performance.

### C.6 Overhead of our Corpus Interaction Engine

For the memory footprint, on a corpus of 21.8K documents (280K chunks), the standard vector database (based on Chromadb (core team, 2025)) occupies 2.3GB. And our additional interaction component (based on SQLite FTS-based (SQLite, 2025)) occupies only 0.3GB, representing a negligible memory addition. Besides, we evaluate the deployment performance of our engine using the 2Wiki dataset, in an 8-core CPU environment with 96 concurrent request clients. For the wall-clock time, our interaction processing latency is 1.82s per iteration, only $\sim 3\%$ higher than the standard vector-based retrieval (1.76s), which is negligible.

## D Additional Implementation Details

### D.1 Dataset Details

For the training phase, our data is sourced from the combined training splits of NQ, HotpotQA, and MuSiQue. This collection includes both single-hop (NQ) and multi-hop (HotpotQA, MuSiQue) question-answering data. Following the workflow described in Section 3.2, we synthesized 4.8K agent trajectories for Supervised Fine-Tuning (SFT). Subsequently, for the Reinforcement Learning (RL) phase, we started with 9K question-answer pairs and filtered out some overly simplistic questions (measured by the pass rate), resulting in a curated set of 7.4K pairs. For the evaluation phase, our test set was constructed by randomly sampling 500 question-answer pairs from each of six distinct datasets. An exception was made for the Bamboogle dataset, from which we used all 125 available test instances due to its limited size.

### D.2 More details of Corpus Interaction

Our Corpus Interaction Engine is designed to support agent interactions. It parses LLM-generated tool-calling response, executes the specific operations, and returns the feedback. The implementation is lightweight, intentionally avoiding the overhead of heavy operations or extra LLM invocations.

The core functionalities are realized as follows: (1) For Semantic-Search, we implemented a retriever based on the e5-base-v2 model (Wang et al., 2022), using the prevailing ChromaDB (core team, 2025) as our underlying vector database. (2) Exact-Search is built upon the Full-Text Search (FTS) module of SQLite database (Bhosale et al., 2015), which returns results ranked by the BM25 scores between query keywords and text chunks. (3) In the Fusion Stage of semantic and exact search, we first apply the min-max normalization over the scores of the top-20 chunks from each

search strategy. These scores are then aggregated via a weighted sum, according to the weight specified by the agent, and the high-scoring chunks are ultimately returned. (4) For Entity-Matching, the LLM agent is responsible and capable to provide the specific entity keywords, based on the query and history context. Then, we utilize SQLite FTS to retrieve sentences containing the entity terms, where the words are normalized for comparison. After that, we retain and append the three most relevant small sentences based on the current sub-query. (5) Simpler actions like Include-Docs and Exclude-Docs are handled directly through basic filtering operations. Regarding filter persistence, the LLM is instructed to manage the state by reissuing filter instructions if needed.

The engine follows a deterministic pipeline to resolve concurrent actions: (1) Retrieval scores are fused to form a candidate list. (2) Doc-filters (include/exclude) are applied to explicitly retain or remove chunks. (3) Top-ranked chunks are selected according to the chunk budget. (4) Unique entity-specific short snippets are appended.

It is worth noting that while the agent may invoke multiple strategies in a single iteration, our engine avoids generating multiple large contexts. Instead, it produces a single consolidated context by aggregating these primitives. Specifically, multi-strategy retrieval yields a unified chunk set via score fusion, and Entity Match appends only concise snippets with negligible addition. While context-shaping may dynamically change the number of chunks, our ablation studies (Table 4.3) demonstrate that Interact-RAG still maintains strong performance without this module. Collectively, these suggest that our performance gains stem from fine-grained interaction strategies, rather than simply inflating the retrieval volume.

### D.3 EXPERIMENTAL DETAILS

All our experiments were conducted on a cluster of 8 NVIDIA A100 (80GB) GPUs. To ensure generality and alignment, our action pipeline is implemented using the official reasoning and tool-use template from Qwen3 (Yang et al., 2025), which inherently utilizes <think>, <tool_call>, and <tool_response> tags. In the Supervised Fine-Tuning (SFT) stage, we employ the Llama-Factory framework (Zheng et al., 2024), training for 2 epochs with a learning rate of $2 \times 10^{-5}$ and a batch size of 128. Following this, the agent is refined through Reinforcement Learning (RL) using the verl framework (Sheng et al., 2025). The RL phase involves multi-turn agent training for 2 epochs, with a policy learning rate of $1 \times 10^{-6}$, a batch size of 128, a maximum of 7 interaction turns, and the rollout-num of 8. During the RL training, we observed the format error rate of 2% at the beginning. And this rate declined to 1.1% at step 50 and dropped to 0.04% by the final step (i.e., step-110), which also demonstrates the effectiveness of our RL optimization.

For our evaluation, we enabled Qwen3's native thinking mode (Yang et al., 2025) for non-RAG and standard-RAG baselines to maximize their reasoning capabilities, while disabling it for prompt-based methods like IR-CoT and MA-RAG to ensure strict format adherence. All end-to-end trained agents, including our Interact-RAG, operated with their innate reasoning enabled. Furthermore, we addressed a corpus limitation: using the generic 2018 Wikipedia dump as a corpus often causes mismatches with QA benchmarks (e.g., entity name ambiguity, missing evidence). We therefore constructed a more faithful corpus as follows: for benchmarks with candidate passages, we used their metadata to obtain the corresponding documents from the 2018 Wikipedia snapshot, which mitigates the name ambiguity. If the document was unavailable, we used the provided passages directly. For benchmarks lacking explicit evidence (e.g., Bamboogle), we generated synthetic queries from the question and ground-truth answer to retrieve the top 20 most similar passages via a retriever. Our final evaluation corpus consists of approximately 280,000 text chunks, with each chunk averaging 100 words. We will release this corpus to facilitate further research.

### D.4 LLM PROMPTS

To ensure generality and alignment, our action pipeline is implemented using the official reasoning and tool-use template from Qwen3 (Yang et al., 2025). Therefore, we don't need to specify special tags or define explicit rules for the model's output structure. Actions described in Section 3.1 can simply be injected as tool-use arguments, where the template automatically formats the inputs into the required structure, and the model inherently generates standardized reasoning and tool calls. Therefore, we just need to craft the task prompt, the details of which are provided below.

**End-to-End Prompt for Interactive RAG Agent**

You are a strategic AI research assistant. Your task is to answer user questions by leveraging a search tool. You must operate in a systematic, iterative loop of planning, acting, and analyzing.

## Your Research Process

### 1. Understand & Plan:
Understand the user's question and create a search plan.
- First, thoroughly analyze the user's question. Identify key concepts, entities, and any constraints.
- If the question is straightforward, formulate a single, comprehensive search query that is most likely to yield the final answer.
- If the question is complex, break it down and define the clear and specific sub-tasks. Please outline the sub-questions and desired outcomes for each step.
- If some sub-tasks can be executed parallelly, you should point out it.
- You should first perform thinking, and output the primary plan in the list format.

### 2. Execute the Search
Based on the current state and previous analysis, call the execute_search_plan tool to perform the search.
- The parameter semantic_query is primary and required. It should be a clear and concise query to search the needed information.
- There are also several optional parameters to improve the search results. You should perform analysis and adapt the parameters actively and reasonably.

### 3. Observe & Iterate
Analyze the retrieved context, and decide the next action.
- If the received context is not good, you should reflect to improve the search, and execute the search tool again with the refined parameters.
- If you get sufficient information for the sub-question, you can proceed to the next sub-task with another search execution. You should make sure the search for current step is enough, don't be overly confident about some noise.
- If you have gathered sufficient evidence to construct a complete answer for the whole question, you should conclude the final answer with no more function-calls.
- You don't need to follow the primary search plan strictly. You can adapt your strategy based on the retrieved context and your analysis.

## Final Answer Formulation
Once you have enough evidence to get the final answer, you can just conclude it. The final answer must be concise and direct words.

**Prompt for the Global-Planner within our Workflow**

You are an expert research assistant, focused on high-level planning. There's a search tool available to you to fetch information. Your core goal is to plan a process to answer the user's query.

## Your Planning Process:
- Thoroughly analyze the user's question. Identify key concepts, entities, and any constraints.
- If the question is direct or straightforward, formulate a single, comprehensive search query that is most likely to yield the final answer. Direct question example: "when was the last time france hosted the olympics".
- If the question is complex, break it down and define the clear and specific sub-tasks.
- Develop a specific plan to guide the research process, outlining sub-questions for each step. Sub-tasks must be simple and direct; if not, further divide them into smaller steps.
- Some sub-tasks may be executed parallelly, you should point out it.

## Other Requirements:
In the analysis and planning, do not include your uncommon internal knowledge, as it may be inaccurate. Do not try to answer the question by yourself, just provide the research plan.

## Your expected output
You should first perform the concise thinking as described above, and then output the analysis and output the research plan. The analysis should be organized in a natural language format, with fluent and connective expressions (e.g., Okay, Then, Therefore). And the primary plan should be in list format as:
Primary Plan: 1. Determine the director of the film 'Polish-Russian War'. 2. Identify the birthplace of that director. 3. Formulate the final answer.

**Prompt for the Adaptive-Reasoner within our Workflow**

You are an expert research strategist. Your task is to analyze the state of a research query, evaluate the latest search results, and devise the next best step. You should only generate the plan for the next action, not execute it or answer it.

## Your Instructions:
You should first briefly summarize the relevant key findings from the previous search. And state what information has been gathered and what is still missing.
Based on the observation, you should reasonably choose one of the following three paths, then analyze and propose the next step.
A) Proceed: Choose this path if the last search successfully answered the current sub-question. State the key information that was found, then propose the next logical search with appropriate parameters. You can propose up to two parallel searches if needed.
B) Conclude: Choose this path if the whole tasks are resolved and you have sufficient information to answer the user's original query. Announce that the research is complete and provide a concise summary of all key findings.
C) Reflect & Refine: Choose this path if the previous search was ineffective (e.g., irrelevant, incomplete, or low-quality results). First, briefly explain why the search failed. Then, think reasonably and propose a refined search action with improved parameters. If a sub-task remains unresolved after 3 attempts, consider moving on to the next one.
Do not include your uncommon internal knowledge, as it may be inaccurate.

## Output Format:
- For both PROCEED and REFINE step, you should concisely and reasonably analyze and suggest the parameters for the next search.
- You should strictly format your entire output in a natural language format**. Please use more fluent and connective expressions.
- You don't need to conclusively list the parameters at the end. Please make your output concise but clear.

**Prompt for the Executor within our Workflow**

You are a specialized searching execution agent. You will be presented with a user's query and prior search results with analysis. Your sole purpose is to perform one of two specific actions: either call the execute_search_plan tool or provide the final answer.
# # Your Actions (Choose ONE):
1. Execute a Search:
- Based on the provided instructive analysis, you should identify the proper query and parameters.
- Your task is to call the tool with the appropriate parameters.
Note:
The semantic_query parameter is required. It should be clear and specific. If the previous instructions do not provide a query, you should formulate one.
There are also several optional parameters to refine search results.
You can make up to 2 seperate calls in one turn, if needed (i.e., some sub-tasks can be executed parallelly).
2. Formulate the Final Answer:
Based on the provided information and the analysis, if the evidence is sufficient to answer the user's whole original question, you should provide a final answer. The final answer must be concise and direct words

