# OpenReview forum: "Interact-RAG: Reason and Interact with the Corpus, Beyond Black-Box Retrieval"
_ICLR.cc/2026/Conference — ICLR 2026 Poster_

### Official Review · Reviewer_Xirv · 2025-10-29

**Soundness:** 3
**Presentation:** 3
**Contribution:** 2
**Rating:** 6
**Confidence:** 4

**Summary:**

The paper introduces Interact-RAG, a paradigm that addresses the black-box limitation in agentic Retrieval-Augmented Generation by granting the LLM agent direct, fine-grained control over the retrieval process through a lightweight Corpus Interaction Engine. This engine provides three categories of action primitives: Multi-Faceted Retrieval, Anchored Matching, and Context Shaping. To enable effective use, the authors propose a modular reasoning-enhanced workflow with a global planner, an adaptive reasoner for proceed and reflect-refine decisions, and an executor, which supports zero-shot execution and synthesizes clean trajectories. These trajectories are used to train an end-to-end agent through Supervised Fine-Tuning on a collection of successful traces, followed by Group Relative Policy Optimization reinforcement learning with a validity-gated accuracy reward. On six benchmarks, the Qwen3-8B-based Interact-RAG achieves higher exact match and F1 scores than existing baselines, showing substantial relative gains and particularly strong improvements on multi-hop reasoning tasks.

**Strengths:**

- The global-planner → adaptive-reasoner → executor decomposition produces coherent traces without verbose LRM noise.
- Consistent SOTA across in- and out-of-distribution.
- lightweight engine adds negligible overhead while enabling actions like adjust_scale for sub-task scoping.

**Weaknesses:**

- Experiments use only 2018 Wikipedia (E5 retriever, top-3 chunks default); no evaluation on larger/dynamic corpora (e.g., 2025 dumps) or non-Wiki domains, where FTS indexing might scale poorly or entity_match fail on ambiguous entities.
- The outcome-only reward ignores process quality—e.g., penalizes all invalid formats equally but doesn't credit efficient trajectories (fewer steps) or penalize over-reliance on exclude_docs, potentially encouraging brittle strategies not exposed in the 7.1K RL questions.

**Questions:**

- What is the per-primitive ablation (EM/F1 drops when removing each category) on HotpotQA and MuSiQue, to quantify if anchored matching alone closes the gap to Search-R1?
- Why mask retrieved tokens only during SFT loss but not RL—does this cause distribution shift, and how many RL trajectories were invalid due to format errors?

---

> ### Author Response · Authors · 2025-11-22
> **Official Reply to Reviewer Xirv [Part 1/2]**
>
> We sincerely thank Reviewer Xirv for the constructive feedback and insightful comments. And we carefully address these comments below.
>
> > **W1.** Evaluation on larger/dynamic corpora (e.g., 2025 dumps) or non-Wiki domains.
>
> Thank you for your valuable comments. We appreciate your suggestion to evaluate our method on more diverse corpora.
>
> Regarding the usage of 2018-Wikipedia dump, we follow the standard and prevailing settings in recent works [1,2,3,4]. Besides, this dump also aligns with the temporal context of the established QA datasets (e.g. HotpotQA), reducing potential knowledge conflicts in newer dumps.
>
> For more comprehensive evaluation over non-wiki domains, we conducted additional experiments on the MultiHop-RAG benchmark [5]. Its corpus consists of news covering diverse fields such as technology, health, and business. As shown in the tables below, Interact-RAG significantly outperforms other baselines in this out-of-distribution setting. This further demonstrates the effectiveness and robustness of our interaction-reasoning paradigm.
>
> We have added these results in the revised version (Appendix C.2).
>
> |                 | EM   | F1   |
> | --------------- | ---- | ---- |
> | Std-RAG-7B      | 57.5 | 59.4 |
> | Std-RAG-8B      | 63.4 | 65.1 |
> | MA-RAG-7B       | 50.2 | 52.3 |
> | MA-RAG-8B       | 66.3 | 68.7 |
> | Search-R1-7B    | 73.0 | 74.1 |
> | Interact-RAG-7B | 79.4 | 80.4 |
> | Interact-RAG-8B | 81.4 | 82.3 |
>
>
>
> > **W2.** The outcome-only reward ignores process quality; concerns about the agent strategies
>
> Thank you for your valuable comments. We appreciate the points regarding the outcome-based reward design.
>
> First, we fully agree that the outcome-only reward may ignore process quality. However, employing GRPO with outcome supervision has become a prevailing method in recent reasoning literature [1] [6] [7]. Following this, we adopt the standard and reliable outcome reward, which encourages the agent to flexibly explore diverse solutions while reducing potential manual bias. Besides, we would like to respectfully clarify that our primary contribution lies in introducing the new Interact-RAG paradigm. Developing an advanced RL algorithm is valuable but beyond the scope of our current work. We have stated this limitation (in Appendix B) and will explore granular rewards and strategies in future work.
>
> Second, our empirical results may alleviate concerns about the policy. Figure 3 demonstrates that Interact-RAG achieves higher accuracy with fewer interaction iterations, and Figure 5 shows that reliance on doc-shaping (e.g., exclude_docs) actually decreases after RL. The generalization on above out-of-distribution benchmark further confirms the robustness of the agent's policy.
>
>
>
> *References:*
>
> > [1] Search-R1: Training LLMs to Reason and Leverage Search Engines with Reinforcement Learning, Jin et al., COLM, 2025
> >
> > [2] R-Search: Empowering LLM Reasoning with Search via Multi-Reward Reinforcement Learning, Zhao et al., arxiv preprint, 2025
> >
> > [3] C-3PO: Compact Plug-and-Play Proxy Optimization to Achieve Human-like Retrieval-Augmented Generation, Chen et al., ICML, 2025
> >
> > [4] InstructRAG: Instructing Retrieval-Augmented Generation via Self-Synthesized Rationales, Wei, et al., ICLR, 2025
> >
> > [5] MultiHop-RAG: Benchmarking Retrieval-Augmented Generation for Multi-Hop Queries, Tang, et al., COLM, 2024
> >
> > [6] DeepSeek-R1: Incentivizing Reasoning Capability in LLMs via Reinforcement Learning, DeepSeek-Team, arxiv preprint, 2025.
> >
> > [7] General-Reasoner: Advancing LLM Reasoning Across All Domains, Ma et al., NeurIPS, 2025.

---

> ### Author Response · Authors · 2025-11-22
> **Official Reply to Reviewer Xirv [Part 2/2]**
>
> > **Q1.** What is the per-primitive ablation? if anchored matching alone closes the gap to Search-R1?
>
> Thank you for your constructive suggestions. To clarify the contribution of each module, we perform additional ablation studies. Specifically, we examined the impact of the interaction classes stated in Section 3.1, by removing each component. The results (EM scores) are as follows:
>
> |                             | 2Wiki | Musique | PopQA |
> | --------------------------- | ----- | ------- | ----- |
> | Interact-RAG-8B                | 69.6  | 34.8    | 56.0  |
> | w/o All-Interaction         | 63.4  | 30.1    | 50.2  |
> | w/o Multi-faceted Retrieval | 66.0  | 34.6    | 55.1  |
> | w/o Anchored Matching       | 66.3  | 34.4    | 53.4  |
> | w/o Context Shaping         | 68.8  | 33.6    | 55.2  |
>
> From the results, we find that:
>
> - **Synergistic Effect:** Removing each single module leads to a performance drop. However, no single removal causes a significant drop as "w/o All-Interaction." This indicates that the primitives are synergistic. They function as a robust collection to support the interactive paradigm and surpass the black-box retrieval. Therefore, the anchored-matching alone does not account for the full performance improvement
> - **Robustness across Patterns:** The contributions of different components vary across data patterns, confirming the comprehensiveness of our design.  For instance, Multi-Faceted Retrieval and Anchored Matching are critical for 2Wiki dataset, which sometimes demands the explicit facts. Context Shaping is more impactful for Musique dataset, where the prevalence of distractors requires robust context refinement.
>
> We have also added these results in our revised version (Section 4.3).
>
>
>
> > **Q2.** Why mask retrieved tokens only during SFT loss but not RL—does this cause distribution shift, and how many RL trajectories were invalid due to format errors?
>
> Thank you for your insightful comments. First, we sincerely apologize for the ambiguity. To clarify, we **did mask** the retrieved tokens during the RL phase, consistent with the SFT stage. This approach ensures that the model focuses on reasoning and action generation, preventing distraction and distribution shift.
>
> Regarding the invalid trajectories due to format errors, we observed an error rate of 2% at the beginning of RL (benefiting from the foundation established in the SFT phase). As training progressed, this rate declined to 1.1% at step 50 and further dropped to 0.04% by the final step (i.e., step 110). These statistics could demonstrate the stability and effectiveness of our RL optimization.
>
> We have revised the paper to further clarify the above content (line 251 and Appendix D.3).

---

> ### Author Response · Authors · 2025-11-28
>
> Dear Reviewer Xirv,
>
> We sincerely appreciate the time you have dedicated to reviewing our paper.
>
> In response to your constructive feedback, we have provided additional experiments and further clarifications. As the discussion period is nearing its end, we would like to know whether our responses have addressed your concerns. If there are remaining questions or suggestions, please do not hesitate to contact us.
>
> Thank you again for your efforts and valuable comments!
>
> Best regards,
>
> Authors

---

### Official Review · Reviewer_DdfH · 2025-10-31

**Soundness:** 4
**Presentation:** 3
**Contribution:** 3
**Rating:** 6
**Confidence:** 4

**Summary:**

Interact-RAG introduces a new paradigm for Retrieval-Augmented Generation that breaks the black-box treatment of retrieval and turns the LLM agent into an active manipulator of the corpus. It provides a lightweight Corpus Interaction Engine with fine-grained action primitives—multi-faceted retrieval (semantic, exact, weighted fusion), anchored matching (entity-focused search), and context shaping (include/exclude docs, adjust scale)—and pairs this with a reasoning-enhanced workflow (global planner, adaptive reasoner, executor) for robust zero-shot execution and data synthesis. The synthesized trajectories are used to train an end-to-end agent via supervised fine-tuning, then refined with reinforcement learning (GRPO). Across six QA benchmarks, Interact-RAG achieves state-of-the-art EM/F1, with especially strong gains on multi-hop tasks and improved efficiency (fewer action iterations). Ablations show the interaction primitives, SFT, and RL are all critical, and the training-free workflow alone outperforms strong multi-agent baselines.

**Strengths:**

Originality: Proposes a corpus interaction engine with actionable primitives (multi-faceted retrieval, anchored matching, context shaping) and a hierarchical reasoning workflow (planner–reasoner–executor), moving beyond black-box query reformulation. The integration of training-free trajectory synthesis with SFT+RL is thoughtfully designed.

Quality: Empirical evaluation is thorough, covering diverse datasets, strong baselines, and ablations that isolate each component’s contribution. Analyses of efficiency (action iterations) and RL training dynamics add credibility. Implementation choices are pragmatic (lightweight FTS, clear tool-call formats).

Clarity: Motivation and problem framing are clear; the pipeline and action space are well explained; figures and tables help understanding; methodology is presented with enough detail to reproduce key components.

Significance: Demonstrates sizable, consistent gains—especially on complex multi-hop tasks—showing the practical value of interactive retrieval. The paradigm is likely to influence future RAG agent design, with both training-free and trained variants offering immediate utility.

**Weaknesses:**

Scalability and deployment realism: The interaction engine relies on SQLite FTS and simple filters. This keeps things lightweight but raises questions about performance on large or sharded corpora, multi-tenant settings, and streaming updates. There is no wall-clock or throughput/latency analysis, nor memory/compute footprint or cost per question. Without these, claims about efficiency (fewer iterations) are not tied to practical runtime advantages.

Retrieval quality metrics: The paper focuses on EM/F1. For multi-hop QA (e.g., HotpotQA), retrieval-oriented metrics (recall@k of supporting docs, precision of included context, coverage of hops) would strengthen the evidence that the interaction engine improves retrieval quality, not just final answers.

Score fusion calibration: Weighted fusion between dense and sparse scores is central, but the paper does not detail score normalization/calibration (dense cosine vs. BM25/FTS scores are not directly comparable), how weights are chosen/updated by the agent, or whether any safeguards (e.g., monotonicity, bounds, or learned scaling) are used. This may lead to unstable retrieval quality across corpora.

Evaluation fairness and backbone effects: The main results use Qwen3-8B while several baselines are reported with 7B checkpoints. Although 7B results for Interact-RAG are in the appendix, fairness is harder to assess from the main text. Bringing same-backbone (7B) head-to-head comparisons into the main results would improve confidence that gains are due to the method rather than model size.

**Questions:**

Fusion mechanics: How are dense and sparse scores normalized before fusion? Are the fusion weights learned, fixed, or chosen by the agent per step? Do you constrain weights (e.g., to [0,1] or sum to 1)? Have you compared your fusion against standard rank fusion methods (RRF, CombSUM/CombMNZ)?

Entity match implementation: How is the “entity” identified and canonicalized (prompted string, NER, or entity linking)? How do you handle aliases, disambiguation, and collisions across similarly named entities? Any safeguards against retrieving wrong-entity passages?

Concurrent actions and precedence: When the agent issues multiple actions at a step (e.g., include/exclude + retrieval), what is the execution order and conflict resolution? How are results aggregated, and how do stateful filters (include/exclude) persist across future steps?

On the “-1 initial penalty” in the reward: In GRPO, if returns are centered/normalized within each group (at least subtracting the group mean, often dividing by the standard deviation), adding a constant to all trajectories (e.g., -1) cancels out and should not affect the advantage or policy updates. Does your implementation include any components that depend on absolute returns (e.g., uncentered advantages, value-function training on absolute returns, cross-group thresholds for filtering/early stopping, running-mean normalization) that would make this bias meaningful? If not, could the -1 be removed or replaced with a more discriminative shaping reward (e.g., graded penalties for format errors, intermediate retrieval-quality rewards), and can you report the impact on training stability and sample efficiency?

---

> ### Author Response · Authors · 2025-11-22
> **Official Reply to Reviewer DdfH [Part 1/2]**
>
> We sincerely thank Reviewer DdfH for the insightful and constructive comments. We carefully address your points below.
>
> > **W1.** Scalability and deployment realism of the interaction engine (based on SQLite FTS and simple filters). Need more latency and memory cost.
>
> Thank you for the valuable feedback. We would like to report the deployment performance of our interaction engine:
>
> - **Memory Footprint**: For our corpus of 21.8K documents (280K document chunks), the standard vector database (based on chroma-db) occupies 2.3GB, and our additional component (based on SQLite FTS) occupies only 0.3GB. This is a marginal addition.
> - **Computational Overhead**: We evaluated the engine processing latency on the 2Wiki dataset, within an 8-core CPU environment. We deploy 96 concurrent request clients, and report the wall-clock retrieval latency. As shown in the table below, our processing latency per iteration (1.82s) is only ~3% higher than the standard vector-based retrieval (1.76s), which is negligible.
>
> |                 | EM score | Retrieval Time (s) / Query | Retrieval Iterations | Time (s) / Iteration |
> | --------------- | -------- | -------------------------- | -------------------- | -------------------- |
> | MA-RAG-7B       | 39.6     | 5.09                       | 2.89                 | 1.76                 |
> | Search-R1-7B    | 50.9     | 5.61                       | 3.16                 | 1.77                 |
> | Interact-RAG-7B | 63.6     | 4.67                       | 2.57                 | 1.82                 |
>
> Besides, we have clarified our terminology, using "retrieval iterations", instead of the "efficiency" to describe more precisely (line 445). And we have added the above results in the revision (Appendix C.6).
>
>
>
> > **W2.** Retrieval quality metrics: The paper focuses on EM/F1. For multi-hop QA, retrieval-oriented metrics would strengthen the evidence that the interaction engine improves retrieval quality, not just final answers.
>
> Thank you for the constructive suggestion. Accordingly, we evaluated the coverage of supporting facts across three multi-hop benchmarks, as they provide human-annotated evidence. We calculated the proportion of supporting facts covered within the retrieved context.
>
> As shown in the table below, our Interact-RAG consistently outperforms other baselines. This confirms that our active interaction paradigm effectively improves the information gathering process. We have included these results in the revised version (Appendix C.3).
>
> |                    | 2Wiki | Musique | HotpotQA |
> | ------------------ | ----- | ------- | ------ |
> | Standard-Retrieval | 0.434 | 0.127   | 0.502  |
> | MA-RAG-7B          | 0.620 | 0.256   | 0.573  |
> | MA-RAG-8B          | 0.631 | 0.277   | 0.581  |
> | Search-R1-7B       | 0.608 | 0.256   | 0.609  |
> | Interact-RAG-7B    | 0.686 | 0.316   | 0.632  |
> | Interact-RAG-8B    | 0.706 | 0.335   | 0.637  |
>
>
>
>
>
> > **W3. & Q1.**  Score fusion calibration: Detail score normalization/calibration, how weights are chosen, or whether any safeguards are used. Fusion mechanics: Do you constrain weights (e.g., to [0,1] or sum to 1)? Have you compared your fusion against standard rank fusion methods?
>
>
>
> Thank you for your valuable feedback. Regarding the fusion mechanics, we first apply min-max normalization to both dense and sparse scores, and then perform the weighted aggregation (as we stated in Appendix D.2). The LLM agent will assign a weight $w$   for the  sparse results, while the dense weight is automatically set to $1-w$. We enforce a safeguard constraint where $w \in [0,1]$  to guarantee stability.
>
> |  | 2Wiki | Musique | PopQA |
> | -------------- | ----- | ------- | ----- |
> | RRF            | 68.7  | 34.6    | 55.3  |
> | CombMNZ        | 69.8  | 32.9    | 56.0  |
> | Ours  | 69.6  | 34.8    | 56.0  |
>
> Following your suggestion, we compare our fusion approach with standard methods and report the results (EM score). As shown in the table, our implementation is robust and slightly superior. Besides, we wish to clarify that the fusion mechanism is a small functional design in our interaction engine, so its variations do not significantly alter the overall performance. Rather than optimizing this specific operator, our design focus remains on the new interaction-reasoning paradigm.
>
> We have added these to the revised version (Appendix C.5).

---

> ### Author Response · Authors · 2025-11-22
> **Official Reply to Reviewer DdfH [Part 2/2]**
>
> > **W4.**  Evaluation fairness and backbone effects: Although 7B results for Interact-RAG are in the appendix, fairness is harder to assess from the main text. Bring the comparisons into the main results.
>
> Thanks for your constructive suggestion. We have moved these results from appendix to the main results (Table 1) in our revised paper. As the results shown, Interact-RAG consistently outperforms others across both backbones, achieving a relative improvement of 22.5% on the 8B and 10.1% on the 7B. These consistent gains confirm that the improvements stem from our interaction-reasoning method.
>
>
>
> > **Q2 & Q3** Entity match implementation: How is the “entity” identified and canonicalized? How do you handle aliases, disambiguation, and collisions across similarly named entities? Any safeguards against retrieving wrong-entity passages?
> > Concurrent actions and precedence: When the agent issues multiple actions at a step (e.g., include/exclude + retrieval), what is the execution order and conflict resolution? How are results aggregated, and how do stateful filters (include/exclude) persist across future steps?
>
> Thank you for your valuable comments. We will clarify more details as follows, and we have also added them in the revised version.
>
> **Entity Match Implementation:** In Interact-RAG, the LLM agent is responsible and capable to provide the specific entity keywords, based on the query and history context. Then, we utilize SQLite FTS to retrieve sentences containing the entity terms, where the words are normalized for comparison. After that, we retain the most relevant sentences based on the current sub-query. This contextual relevance check acts as an implicit safeguard to filter out irrelevant aliases or ambiguous entities. Furthermore, the agent itself acts as a cognitive safeguard: even if erroneous passages are retrieved, the LLM can analyze and detect the mismatch, actively refining the subsequent retrieval.
>
> **Concurrent Actions & Precedence:** The execution follows a deterministic pipeline to resolve conflicts. First, dense and sparse retrieval scores undergo min-max normalization and weighted aggregation to form a top-20 candidate chunk-list. Second, filters (include/exclude) are applied to this list to explicitly retain or remove chunks. After that, the top-ranked relevant chunks are selected according to the chunk budget. Finally, the unique entity-specific short sentences are appended last. Regarding filter persistence, the LLM agent is instructed to manage the state by reissuing filter instructions if needed. This allows the agent to flexibly maintain, adjust or correct its decision during the process.
>
> We have also clarified the above implementation in revision (Appendix D.2).
>
>
>
> > **Q4.** On the “-1 initial penalty” in the reward: In GRPO, if returns are centered/normalized within each group, adding a constant to all trajectories cancels out and should not affect the advantage or policy updates.
>
> Thank you for your insightful comments. We fully agree with your analysis.
>
> The inclusion of the "-1 initial penalty" was primarily intended for intuitive logic. It ensures that trajectories with format errors explicitly yield negative values (-1), matching the "invalid penalty". Our implementation relies on standard relative advantages, not dependent on absolute return values. Mathematically, as you correctly pointed out, this constant term is canceled out during the group-based normalization process in GRPO and does not influence the advantages or the actual training.  Besides, as demonstrated in Figure 4, our reinforcement learning with this reward can yield stable training and convergence.
>
> We appreciate this clarification and explicitly state this in the revised Section 3.3 (line 275).

---

> ### Author Response · Authors · 2025-11-28
>
> Dear Reviewer DdfH,
>
> We sincerely appreciate the time you have dedicated to reviewing our paper.
>
> In response to your constructive feedback, we have provided additional experiments and further clarifications. As the discussion period is nearing its end, we would like to know whether our responses have addressed your concerns. If there are remaining questions or suggestions, please do not hesitate to contact us.
>
> Thank you again for your efforts and valuable comments!
>
> Best regards,
>
> Authors

---

### Official Review · Reviewer_6cgB · 2025-10-31

**Soundness:** 3
**Presentation:** 3
**Contribution:** 2
**Rating:** 4
**Confidence:** 4

**Summary:**

The paper proposes Interact-RAG, which exposes retrieval as controllable “interaction primitives” (semantic/exact search, weighted fusion, entity match, context shaping) instead of a black-box query. A reasoning-enhanced workflow (planner–reasoner–executor) is used for zero-shot execution and data synthesis, followed by SFT and RL to train an end-to-end agent. Experiments on six QA benchmarks show sizable gains over agentic baselines.

**Strengths:**

1. This paper has practical significance in addressing a key limitation in agentic RAG—namely, the lack of retrieval control—making it potentially useful for real-world systems.

2. The paper is clearly written and easy to follow, with well-structured explanations that make the methodology and experiments accessible.

**Weaknesses:**

1. The proposed Corpus Interaction Engine appears incremental rather than fundamentally novel; it mainly uses additional retrieval modes. This does not fully realize the claim in intro section:  “transforming the agent from a passive query issuer to an active participant,”.

2. Missing fine-grained ablations: Ablation studies for the specific components (Multi-Faceted Retrieval, Entity Match, Adjust Scale, and Doc Shaping) are absent, making it unclear which modules drive the observed performance gains and by how much.

3. Backbone and retriever parity: Please report results with matched backbones (same LLM family/size and the same retriever). The paper uses e5-base-v2—what retrievers are used for each baseline, and are they consistent? Without parity, improvements may reflect model or retriever differences rather than the proposed method.

4. Cost analysis: Because the method’s reasoning trajectory introduces additional steps, the extra wall-clock time and token costs should be analyzed in detail (end-to-end latency, tool-call time, and total token consumption).

5. Retrieval step inflation: From Figures 5 and 6, a single output trajectory in Interact-RAG can issue multiple tool_call operations, and both Multi-Faceted Retrieval and Entity Match can trigger several retrieval steps. Compared to Search-R1 and other baselines, how many retrieval steps are taken in total? Could the performance gains be primarily due to increased retrieval volume rather than a better strategy? Please quantify and control for the number of retrievals.

**Questions:**

Please address the concerns in Weaknesses.

---

> ### Author Response · Authors · 2025-11-22
> **Official Reply to Reviewer 6cgB  [Part 1/2]**
>
> We sincerely thank Reviewer 6cgB for the insightful review and the constructive comments. Below, we carefully address your comments point by point.
>
> > **W1.** The proposed Corpus Interaction Engine appears incremental rather than fundamentally novel; it mainly uses additional retrieval modes. This does not fully realize the claim in intro section: “transforming the agent from a passive query issuer to an active participant
>
> Thank you for your valuable comments. We would like to respectfully clarify that Interact-RAG’s novelty lies in transforming the control flow of retrieval, more than the diversity of retrieval modes. Existing agentic RAG methods operate in a passive trial-and-error loop, blind-guessing new queries without visibility into the black-box retrieval process [1] [2] [3]. In contrast, our Interact-RAG enables **active manipulation**, where the agent can proactively analyze, diagnose and control the retrieval process with fine-grained primitives (e.g., focus more on specfic entities, exclude the noise).
>
> The Corpus Interaction Engine provides these primitives to **primarily support** the manipulation, and our **data-synthesis pipeline and 2-stage training** **further empower** the LLM for active and strategical control. These **collectively** represent a shift from trial-and-error query guessing to reasoning-driven active retrieval management. We've further clarified this in the introduction section.
>
>
>
> > **W2.** Additional ablation studies for the specific components.
>
> Thank you for your constructive suggestions. To clarify the contribution of each module, we perform additional ablation studies. Specifically, we examined the impact of the interaction classes stated in Section 3.1, by removing each component. The results (EM scores) across three datasets are as follows:
>
> |                             | 2Wiki | Musique | PopQA  |
> | --------------------------- | ----- | ------- | ---- |
> | Interact-RAG-8B            | 69.6  | 34.8    | 56.0 |
> |   w/o All-Interaction         | 63.4  | 30.1    | 50.2 |
> |   w/o Multi-faceted Retrieval | 66.0  | 34.6    | 55.1 |
> |   w/o Anchored Matching       | 66.3  | 34.4    | 53.4 |
> |   w/o Context Shaping         | 68.8  | 33.6    | 55.2 |
>
> From the results, we find that:
>
> - **Synergistic Effect:** Removing each single module leads to a performance drop. However, no single removal causes a significant drop as "w/o All-Interaction". This indicates that the primitives are synergistic. They function as a robust collection to support the interactive paradigm and surpass the black-box retrieval.
> - **Robustness across Patterns:** The contributions of different components vary across data patterns, confirming the comprehensiveness of our design.  For instance, Multi-Faceted Retrieval and Anchored Matching are critical for 2Wiki dataset, which sometimes demands the explicit facts. Context Shaping is more impactful for Musique dataset, where the prevalence of distractors requires robust context refinement.
>
> We have added these results in our revised version (Section 4.3).
>
>
>
> > **W3.** Backbone and retriever parity.
>
> Thank you for your valuable feedback. We have carefully ensured parity throughout our experiments.
>
> - **Retriever:** We standardized the retrieval setup across all experiments and baselines. We utilized the same e5-base-v2 retriever and the same 2018 Wikipedia corpus **for all methods**. This ensures consistent evaluation, and we have further clarify this in Section 4.1 (line 304).
> - **LLM Backbone:** We use the latest prevailing Qwen3-8B model as the primary backbone for all methods, except for some marked methods (like Search-R1), where we use their official 7B checkpoints since their 8B versions are unavailable. **To further ensure a fair comparison,** we additionally conducted experiments using **Qwen2.5-7B for all methods;** these results were **originally reported in Appendix B.1** (and have been appended to Table 1 in the revision).  Interact-RAG consistently outperforms other baselines across both backbones, achieving a relative improvement of 22.5% on the 8B and 10.1% on the 7B. These consistent gains confirm that the improvements stem from our interaction-reasoning method.
>
>
>
> *References:*
>
> > [1] Search-R1: Training LLMs to Reason and Leverage Search Engines with Reinforcement Learning, Jin et al., COLM, 2025
> >
> > [2] Chain-of-Retrieval Augmented Generation, Wang et al, NeurIPS, 2025
> >
> > [3] R-Search: Empowering LLM Reasoning with Search via Multi-Reward Reinforcement Learning, Zhao et al., arXiv preprint, 2025

---

> ### Author Response · Authors · 2025-11-22
> **Official Reply to Reviewer 6cgB [Part 2/2]**
>
> > **W4.** Cost analysis: end-to-end latency, tool-call time, and total token consumption.
>
> Thank you for the valuable suggestion. We have added a detailed analysis of the computational costs over the 2WikiMultiHopQA dataset, as summarized in the table below.
>
> |                 | EM | Latency (s) | Tool-call Time (s) | Out Tok. | In Tok. |
> | --------------- | -------- | ------------------------------ | -------------------------- | ------------ | ----------- |
> | MA-RAG-7B       | 39.6     | 15.1                          | 5.09                      | 181          | 3812        |
> | Search-R1-7B    | 50.9     | 17.2                          | 5.61                      | 192          | 4289        |
> | Interact-RAG-7B | 63.6     | 36.4                          | 4.67                      | 713          | 5216        |
>
> The time is measured on the NVIDIA A100 GPU with the vLLM engine. We report the
> EM score alongside average wall-clock latency, tool execution time, and
> token consumption per query. Since each RAG processing contains multiple turns, so the token count is accumulated.
>
> We acknowledge that Interact-RAG incurs higher latency and token usage due to its Chain-of-Thought reasoning trajectory. However, this design aligns with the **emerging paradigm of long-CoT reasoning models** and **test-time scaling,** where increased compute is a **necessary trade-off** for superior accuracy over complex tasks [1] [2] [3] [4]. We have clarified this in our limitation section (Appendix B). In furture work, we will further regulate the length of CoT trajectories during the data synthesis and training, optimizing the reasoning efficiency.
>
> Furthermore, we applied **Best-of-N** scaling (N=3) (i.e., sampling and selection) on Search-R1 baseline. While this increases its computational cost by over 3x, the accuracy just improved from 50.9 to 56.2, **still falling short** of ours. This demonstrates the effectiveness of Interact-RAG.
>
> We have included the above cost analysis in the revision (Appendix C.4).
>
>
>
> > **W5.** Retrieval step inflation with multiple tool-call operations.
>
> Thank you for your valuable comments. We appreciate the opportunity to clarify the mechanism of our Engine. We apologize for the potential misunderstanding regarding the processing of multiple tool-calls.
>
> A single retrieval iteration, with multiple tool-call operations, does not result in separate multiple large contexts. Instead, our engine parses the tool-calls and generates a consolidated context. For example, Multi-Faceted Retrieval fuses scores from semantic and exact strategies and returns a single set of chunks. Thus, multiple strategies do not introduce extra chunks. Similarly, the Entity Match only appends three entity-specific short sentences rather than large chunks, adding negligible volume. While context-shaping may dynamically change the number of chunks, our ablation studies (in W2) demonstrate that Interact-RAG still maintains strong performance without this module. Moreover, despite processing multiple operations per iteration, the additional engine cost remains minimal (3%) due to the low overhead implementation (more in Appendix C.6). Furthermore, as shown in Figure 3, our agent can achieve higher accuracy with fewer retrieval iterations.
>
> Collectively, these suggest that our performance gains stem from fine-grained interaction strategies, rather than simply inflating the retrieval volume. We have added this clarification in the revised version (Appendix D.2).
>
>
>
> *References:*
>
> > [1] DeepSeek-R1: Incentivizing Reasoning Capability in LLMs via Reinforcement Learning, DeepSeek-Team, arxiv preprint, 2025.
> >
> > [2] Demystifying Long Chain-of-Thought Reasoning, Yang et al, ICML, 2025
> >
> > [3] Chain-of-Retrieval Augmented Generation, Wang et al, NeurIPS, 2025
> >
> > [4] Metastable Dynamics of Chain-of-Thought Reasoning: Provable Benefits of Search, RL and Distillation, Kim et al, ICML, 2025

---

> > ### Comment · Reviewer_6cgB · 2025-11-26
> >
> > Thanks for the authors’ clarifications. All of my concerns are addressed, and I will raise my score to 6.

---

> > > ### Author Response · Authors · 2025-11-27
> > >
> > > Dear Reviewer 6cgB,
> > >
> > > Thanks for your positive feedback! We are glad to hear that all your concerns have been addressed. We sincerely appreciate the time and effort you have dedicated to reviewing our paper and providing valuable comments.
> > >
> > > Best regards,
> > >
> > > Authors

---

### Official Review · Reviewer_rrcZ · 2025-11-03

**Soundness:** 3
**Presentation:** 3
**Contribution:** 3
**Rating:** 6
**Confidence:** 2

**Summary:**

This paper addresses a critical limitation in contemporary Retrieval-Augmented Generation (RAG) systems, specifically within the "Agentic RAG" paradigm. The central problem identified is that existing LLM agents treat the retrieval process as an "opaque black-box". This confines the agent to the role of a "passive query issuer," which must rely on inefficient "trial-and-error loops" of query reformulation to find information, a method that frequently fails in complex, multi-hop information-seeking tasks.
The paper hypothesizes that empowering the LLM agent with fine-grained, transparent control over the retrieval process will lead to significantly more effective and efficient information seeking. To test this, the authors introduce Interact-RAG, a new paradigm that transforms the agent into an "active manipulator" of the retrieval process.

**Strengths:**

1. The questions posed by this paper are critical to the advancement of LLM agents. The core problem—that agents are "stuck" in inefficient query reformulation loops and lack fine-grained control over their tools —is a widely recognized and significant unsolved challenge in the field. This paper tackles this gap head-on, addressing the fundamental limitations of the agent-retriever interface.
2. The paper is exceptionally rigorous. The design of the Interact-RAG-Workflow as a dual-purpose system—serving as both a strong zero-shot agent and a high-quality data synthesizer—is an elegant and insightful solution to the data bottleneck problem in agent training.
3. The authors' thoroughness is a key strength. The ablation studies (Table 2)  and detailed behavioral analysis (Section 4.4)  anticipate and preemptively answer nearly every major question a reviewer might have about why the system works. The analysis of learned interaction patterns in Figure 5, for instance, provides a fascinating window into the "mind" of the trained agent, showing how its policy evolves to favor precision (e.g., Entity-Match).

**Weaknesses:**

1. This is the paper's most significant weakness. The authors admit in Appendix C.3 that the standard 2018 Wikipedia dump has "mismatches" and "missing evidence," leading them to "construct a more faithful corpus". This is a major methodological decision that clouds the results. The paper does not explicitly state that the baselines (Search-R1, R-Search, etc.) were re-evaluated on this new "faithful corpus." If they were not, the 22.5% gain and SOTA claims are confounded, as the comparison would not be apples-to-apples. This corpus advantage could be responsible for a portion of the performance gain.
2. The reward function $R(\tau) = -1 + \mathbb{I}\{\tau_{valid}\} + \mathbb{I}\{\tau_{valid}\} \cdot \mathbb{I}\{y_{ans}\}$ 1 is highly sparse. It only provides a positive signal at the end of a (potentially long) trajectory. The paper does not discuss why this sparse signal was sufficient for the agent to learn complex, multi-step strategies (like prioritizing Entity-Match, as seen in Figure 5).1 The implicit answer, supported by the failure of the RL-only "w/o SFT" agent 1, is that the SFT stage 1 does almost all the heavy lifting. The RL stage merely refines an already-strong policy. The paper's true methodological contribution may be the SFT data-generation pipeline, with the RL refinement being a final optimization step.

**Questions:**

1.  [most urgent point ] Were the baseline models (Search-R1, R-Search, etc.) re-evaluated on the "faithful corpus"?
2. Could the authors please justify or rephrase the "lightweight" claim? Given the architectural requirement for two distinct search indexes (a vector database and an FTS module) , the system is architecturally more complex than standard RAG. Perhaps "computationally efficient" (referencing the fewer iterations in Figure 3)  is a more accurate term than "lightweight" (which implies low architectural overhead)

---

> ### Author Response · Authors · 2025-11-22
> **Official Reply to Reviewer rrcZ**
>
> We sincerely thank Reviewer rrcZ for the thoughtful and constructive feedback. And we carefully address your comments below.
>
> > **W1 & Q1:**  Were the baseline models re-evaluated on the constructed "faithful corpus?
>
> Thank you for your valuable comments. We explicitly confirm that **all baseline models were re-evaluated** on the constructed "faithful corpus" under the exact same setting. The performance gains stem from our Interact-RAG method. The comparison reported in the paper is strictly apples-to-apples, and we have further highlighted this fair comparison in Section 4.1 (line 303).
>
>
>
> > **W2:**  About the reward function and the effectiveness of the RL stage.
>
> We appreciate this insightful observation, and we offer the following clarifications:
>
> - **Validate the reward strategy and our focus.** While the outcome-based reward is sparse, employing GRPO with outcome supervision has become a prevailing and effective practice in recent reasoning and agentic literature [1] [2] [3]. The outcome-reward provides a direct and reliable signal that encourages our LLM agent to flexibly explore diverse solutions and learn to approach to the effective strategies.  Besides, we would like to clarify that our primary contribution lies in introducing the new Interact-RAG paradigm, which dismantles the black-box of traditional retrieval. Developing an advanced RL algorithm is valuable but beyond the scope of our current work; therefore, we adopt an established and standard training recipe to support our paradigm. We have stated this limitation in Appendix B, and will explore granular rewards and strategies in future work.
> - **The Necessity of RL**. We agree that the SFT stage (along with the data-generation) is important. However, we would like to respectfully clarify that the RL training is also a necessary optimization phase.  First, as shown in our ablation (Table 2 and Figure 4), removing the RL stage leads to significant performance drops, which indicates that the SFT policy alone is insufficient. Second, as analyzed in Section 4.4, SFT is crucial for "warm-starting" the model, teaching it the basic patterns; and then the RL largely improves the agent's policy, with notable strategy refinement and shifts (e.g., prioritizing entity-match).
>
> > **Q2:** Could the authors please justify or rephrase the "lightweight" claim of the architecture?
>
> We thank the reviewer for this precise observation. Our use of "lightweight" referred to the small computational overhead of the interaction engine and primitives, where our additional FTS indexing introduces negligible cost (more details in Appendix C.6).  We adopt the reviewer’s suggestion and refine this phrasing in the revised paper (line 187).
>
>
>
> *References:*
>
> > [1] Search-R1: Training LLMs to Reason and Leverage Search Engines with Reinforcement Learning, Jin et al., COLM, 2025.
> >
> > [2] DeepSeek-R1: Incentivizing Reasoning Capability in LLMs via Reinforcement Learning, DeepSeek-Team, arxiv preprint, 2025.
> >
> > [3] General-Reasoner: Advancing LLM Reasoning Across All Domains, Ma et al., NeurIPS, 2025.

---

> ### Author Response · Authors · 2025-11-28
>
> Dear Reviewer rrcZ,
>
> We sincerely appreciate the time you have dedicated to reviewing our paper.
>
> In response to your constructive feedback, we have provided further clarifications. As the discussion period is nearing its end, we would like to know whether our responses have addressed your concerns. If there are remaining questions or suggestions, please do not hesitate to contact us.
>
> Thank you again for your efforts and valuable comments!
>
> Best regards,
>
> Authors

---

### Author Response · Authors · 2025-11-22
**Overall Response**

We sincerely thank all reviewers for their time and insightful comments, which have significantly helped to improve our manuscript.  Here, we provide an overview of the reviewers' comments and outline the main updates we have made.

**The main strengths of our submission:**

We are encouraged that the reviewers have acknowledged the strengths of our work, including:

- Our paper addresses a critical problem in agentic RAG.  [Reviewer rrcZ, 6cgB]

- The proposed design is original and insightful. [Reviewer rrcZ, DdfH]

- Our method achieves consistent and substantial improvement.  [Reviewer 6cgB, DdfH, Xirv]

- The experiments are thorough and comprehensive.  [Reviewer rrcZ, DdfH]

- The paper is clearly written, well-structured, and easy to follow. [Reviewer 6cgB, DdfH]



**Common suggestions and concerns:**

* Add more granular ablation studies.
* Provide more explanation regarding the implementation and RL design.
* Clarify the fairness of the evaluation.



**Our main updates:**

In response to the reviewers' constructive feedback, we have carefully revised our paper and provided additional experiments and clarifications.

* We added detailed ablation studies on specific interaction modules (Section 4.3)
* We provided more details and explanation of the implementation and RL design  (Appendix D.2, Section 3.3)
* We further clarified the fairness of our comparison  (Section 4.1 and line 350)
* We extended the evaluation to non-wikipedia domains (Appendix C.2)
* We provided more experimental analysis on retrieval quality, cost-performance trade-off, fusion policy, and engine overhead. (Appendix C.3, C.4, C.5, C.6)
* We further refined certain expressions for better clarity.

---

### Author Response · Authors · 2025-12-02
**Message to Area Chair**

Dear Area Chair,

We sincerely thank you for managing our submission, especially under the unusual circumstances posed by the recent OpenReview incident. We deeply appreciate your commitment to maintaining a fair review process.

We respectfully wish to point out that our paper achieved a positive consensus through technical discussion **prior to** the data leakage incident.

- **Revision Status:** We have submitted a comprehensive rebuttal that thoroughly addresses all reviewer comments. And we have revised the manuscript to reflect the updates.
- **Score Improvement:** Our initial ratings were **6/6/6/4**. During the discussion phase, Reviewer 6cgB engaged with us and confirmed that all the concerns were addressed. Consequently, the score was **raised from** **4 to 6**.
- **Timeline Integrity:** This score adjustment was finalized on **November 26,**  **24 hours before** the OpenReview leakage. The other three reviewers (who initially gave positive scores of 6) have not provided new feedback.

In summary, our submission reached the ratings of **6/6/6/6**, strictly through fair technical discussion, before the OpenReview leakage. We believe this positive consensus strongly supports the merit of our submission.

Thank you once again for your time and dedication.

Best regards,

The Authors

---

### Meta-Review · Area_Chair_fA34 · 2026-01-01

**Summary:**

This paper proposes Interact-RAG, which makes LLM agents active manipulators of the retrieval process. By introducing a Corpus Interaction Engine with primitives (e.g., Multi-Faceted Retrieval, Context Shaping) and a reasoning-enhanced workflow trained via SFT and RL, the method achieves superior performance on multi-hop QA benchmarks compared to standard agentic RAG baselines.

**Reviewer Concerns:**

The authors comprehensively addressed the majority of concerns, particularly regarding experimental validity and component attribution. The most critical concern regarding the fairness of the "faithful corpus" (Reviewer rrcZ) was resolved by confirming that all baselines were re-evaluated under identical conditions. Requests for granular ablations to isolate the impact of specific interaction primitives (Reviewers 6cgB, Xirv) were met with new data demonstrating the synergistic effect of the modules. Concerns regarding domain generalization (Reviewer Xirv) were addressed by adding the non-Wikipedia MultiHop-RAG benchmark. While the cost/latency trade-off (Reviewers 6cgB, DdfH) remains an inherent characteristic of the method, the authors provided transparent analysis showing the overhead is acceptable given the performance gains.

**Reviewer Scores:**

Reviewer 6cgB explicitly raised their score from 4 to 6 during the discussion, confirming that the clarifications on retrieval step inflation and the addition of component ablations resolved their concerns.

Reviewer rrcZ is likely to maintain the positive score. Their primary hesitation was the "faithful corpus" methodology; the authors' confirmation that baselines were re-evaluated ensures the claims are apples-to-apples.

Reviewer DdfH and Reviewer Xirv are also likely to maintain their scores of 6 or slightly increase. Both initially gave positive ratings, and the rebuttal significantly strengthened the paper by adding out-of-domain experiments (MultiHop-RAG), detailed deployment metrics (memory/latency), and "supporting fact coverage" analysis.

---

### Decision · Program_Chairs · 2026-01-26

Accept (Poster)